# RNA helicase SKIV2L limits antiviral defense and autoinflammation elicited by the OAS-RNase L pathway

Kun Yang [1]✉, Beihua Dong[2], Abhishek Asthana[2], Robert H Silverman [2] & Nan Yan [1]✉

## Abstract

The OAS-RNase L pathway is one of the oldest innate RNA sensing pathways that leads to interferon (IFN) signaling and cell death. OAS recognizes viral RNA and then activates RNase L, which subsequently cleaves both cellular and viral RNA, creating "processed RNA" as an endogenous ligand that further triggers RIG-I-like receptor signaling. However, the IFN response and antiviral activity of the OAS-RNase L pathway are weak compared to other RNA-sensing pathways. Here, we discover that the SKIV2L RNA exosome limits the antiviral capacity of the OAS-RNase L pathway. SKIV2L-deficient cells exhibit remarkably increased interferon responses to RNase L-processed RNA, resulting in heightened antiviral activity. The helicase activity of SKIV2L is indispensable for this function, acting downstream of RNase L. SKIV2L depletion increases the antiviral capacity of OAS-RNase L against RNA virus infection. Furthermore, SKIV2L loss exacerbates autoinflammation caused by human OAS1 gain-of-function mutations. Taken together, our results identify SKIV2L as a critical barrier to OAS-RNase L-mediated antiviral immunity that could be therapeutically targeted to enhance the activity of a basic antiviral pathway.

**Keywords** SKIV2L; RNA Exosome; OAS; RNase L; IFN Response
**Subject Categories** Immunology; Microbiology, Virology & Host Pathogen Interaction; Signal Transduction

## Introduction

The 2',5'-oligoadenylate synthetase (OAS)-RNase L pathway is one of the first characterized innate immune pathway that senses cytoplasmic viral RNA (Kerr and Brown, 1978; Zhou et al, 1993). OAS family genes (*OAS1-3* and *OASL*) are IFN-stimulated genes (ISGs) that encode pattern recognition receptors (PRRs). OAS1-3 sense dsRNA and produce 2',5' linked oligoadenylate (2–5A) from ATP (Kristiansen et al, 2011). 2–5A acts as a second messenger that activates constitutively expressed RNase L. Activated RNase L cleaves cellular and viral RNAs to produce processed RNA species with 2',3'-cyclic phosphate at the 3' end, which is biochemically compatible for activation of RIG-I (Jung et al, 2020; Malathi et al, 2007). RNase L is thought to be the final step of RNA processing before exerting antiviral effector functions.

Although both the OAS-RNase L and RIG-I/MDA5-MAVS pathways sense viral RNA and activate the IFN response, it has long been known that direct sensing of exogenous RNA by RIG-I/MDA5 contributes to most of the IFN production whereas the contribution from sensing RNase L-processed RNA is negligible or minimal. Consistent with this notion, *Mavs*-deficient cells and mice are highly susceptible to a broad spectrum of RNA viruses, whereas *Rnasel*-deficient cells and mice are only moderately susceptible to a few viruses (Chakrabarti et al, 2015; Drappier and Michiels, 2015; Zhou et al, 1997). The IFN-inducible nature of OAS expression and the viral antagonism of this pathway could be limiting factors at the immediate onset of a viral infection. However, IFN strongly induces the expression of OAS proteins that synthesize large amounts of 2–5A that activates a single target, RNase L, which should theoretically produce many more RNA ligands for RIG-I than that of the invading virus. This raises the speculation that the majority of RNase L-processed RNA may be lost before they can activate antiviral immunity.

The RNA exosome is an evolutionarily conserved cellular 3'-5' RNA degradation machinery in eukaryotes, which is involved in RNA processing, maturation, surveillance and turnover (Houseley et al, 2006). In mammalian cells, different RNA exosomes are present in the nucleus, nucleolus or cytoplasm and are associated with different cofactor complexes to target different RNA substrates (Kilchert et al, 2016). The human super-killer complex (SKI), consisting of Ski2-like RNA helicase (SKIV2L), tetratricopeptide repeat domain 37 (TTC37) and WD repeat domain 61 (WDR61) forms the cofactor complex for the cytoplasmic RNA exosome (Halbach et al, 2013). SKIV2L is an RNA helicase that unwinds RNA substrates and threads them through RNA exosome for degradation, while TTC37 and WDR61 contribute to the structure and activity of the SKI complex. The role of SKIV2L and the cytoplasmic RNA exosome in antiviral immunity is unknown. Here, we identify SKIV2L as a critical barrier downstream of RNase L that severely limits the antiviral capacity of the OAS-RNase L pathway.

---

[1]Department of Immunology, UT Southwestern Medical Center, Dallas, TX, USA. [2]Department of Cancer Biology, Cleveland Clinic, Cleveland, OH, USA.
✉E-mail: kun.yang@utsouthwestern.edu; nan.yan@utsouthwestern.edu

# Results

## The SKI complex limits dsRNA-induced innate immune response and apoptosis

To investigate whether SKIV2L plays a role in innate immune response to dsRNA, we generated *SKIV2L* knockout (*SKIV2L*[KO]) A549 cells and stimulated cells with dsRNA analog poly(I:C) (low molecular weight, throughout the study unless specified otherwise)

(Fig. 1A). Two independent clones of *SKIV2L*[KO] cells showed drastically increased (> eightfold) IFN production to poly(I:C) compared to that in wild-type (WT) cells (Fig. 1B). Importantly, stable ectopic expression of wild-type SKIV2L reduced IFN response in *SKIV2L*[KO] cells to a similar level of WT cells (Fig. 1C,D), suggesting that the enhanced IFN response was indeed caused by loss of SKIV2L. As typical of the DExH family of RNA-dependent ATPase, SKIV2L has a helicase region containing an evolutionarily conserved catalytic core (Fig. 1C). We found that SKIV2L E424Q

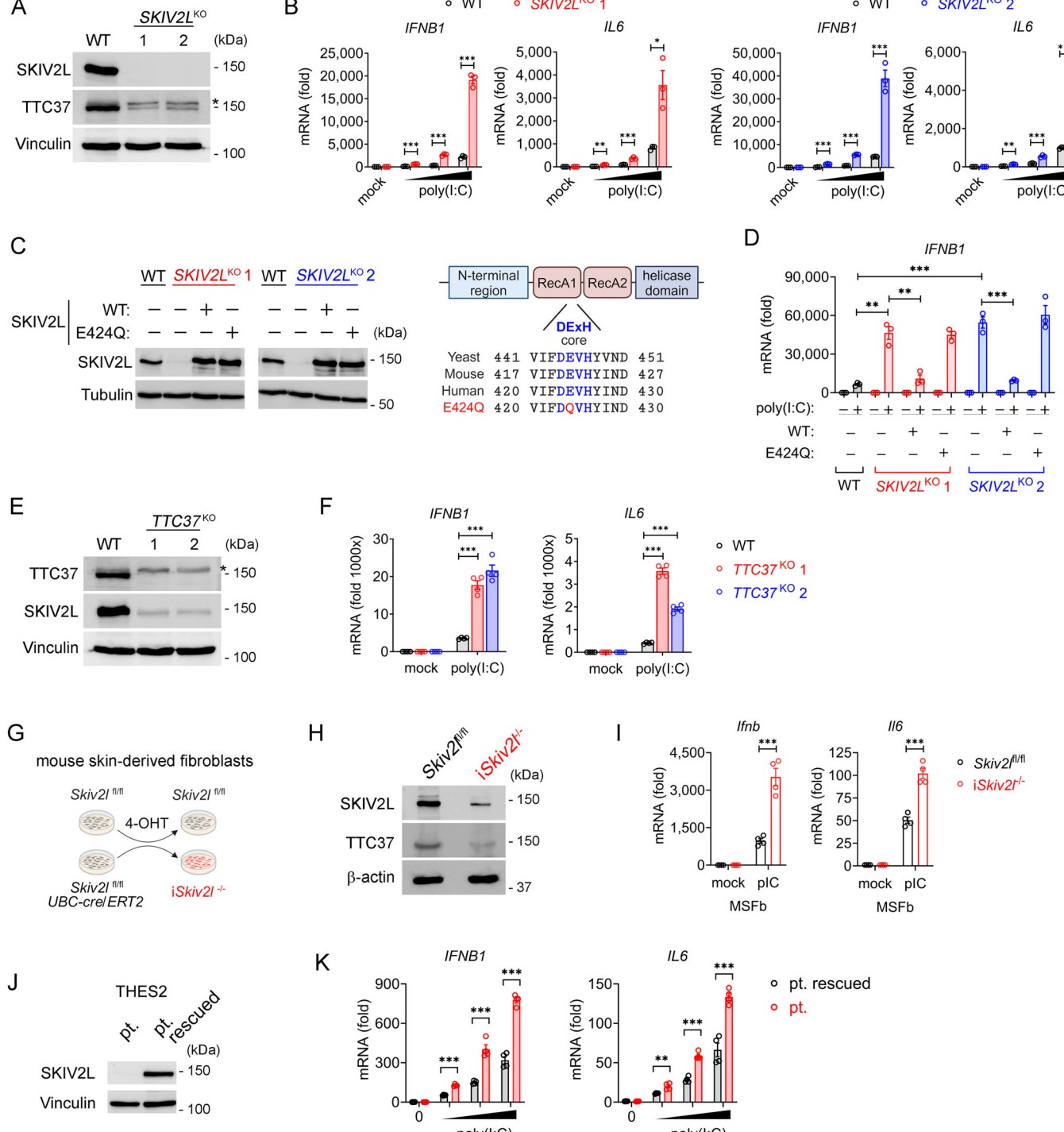

**Figure 1.** *SKIV2L deficiency enhances dsRNA-induced type I IFN response.*

(A) Western blot analysis of SKIV2L and TTC37 proteins in WT and two independent lines of $SKIV2L^{KO}$ A549 cells. *, a non-specific band. (B) RT-qPCR analysis of *IFNB1* and *IL6* mRNA in WT and two independent lines of $SKIV2L^{KO}$ cells treated with increasing dose of poly(I:C) (0.1, 0.3, 1.0 µg/ml) for 4 h. Data are shown as mean ± SEM ($n = 3$ each genotype). Two-sided Student's $t$ test; *$P < 0.05$, **$P < 0.01$, ***$P < 0.001$. $P$ values $= 0.000144$, $0.000069$, $0.000041$ (*IFNB1*, WT vs $SKIV2L^{KO}$ 1 from left to right); $0.005010$, $0.000662$, $0.012617$ (*IL6*, WT vs $SKIV2L^{KO}$ 1 from left to right); $0.000220$, $0.000007$, $0.000712$ (*IFNB1*, WT vs $SKIV2L^{KO}$ 2 from left to right); $0.001757$, $0.000724$, $0.000163$ (*IL6*, WT vs $SKIV2L^{KO}$ 2 from left to right). (C) Western blot analysis of SKIV2L protein in $SKIV2L^{KO}$ cells reconstituted with WT or E424Q mutant SKIV2L. A schematic diagram showing SKIV2L domains and conserved DExH core (right). (D) RT-qPCR analysis of *IFNB1* mRNA in WT, $SKIV2L^{KO}$ cells and $SKIV2L^{KO}$ cells reconstituted with WT or E424Q mutant SKIV2L after poly(I:C) (1.0 µg/ml) treatment. Data are shown as mean ± SEM ($n = 3$ each genotype). Two-sided Student's $t$ test; **$P < 0.01$, ***$P < 0.001$. $P$ values $= 0.001366$ (WT vs $SKIV2L^{KO}$ 1), $0.000396$ (WT vs $SKIV2L^{KO}$ 2), $0.003549$ ($SKIV2L^{KO}$ 1 vs $SKIV2L^{KO}$ 1 + WT-SKIV2L), $0.000487$ ($SKIV2L^{KO}$ 2 vs $SKIV2L^{KO}$ 2 + WT-SKIV2L). (E) Western blot analysis of TTC37 and SKIV2L proteins in WT and two independent lines of $TTC37^{KO}$ cells. *, a non-specific band. (F) RT-qPCR analysis of *IFNB1* mRNA in WT and two independent lines of $TTC37^{KO}$ cells treated with poly(I:C) (1.0 µg/ml) for 4 h. Data are shown as mean ± SEM ($n = 4$ each genotype). Two-sided Student's $t$ test; ***$P < 0.001$. $P$ values $= 0.000020$ (*IFNB1*, WT vs $TTC37^{KO}$ 1), $0.000020$ (*IFNB1*, WT vs $TTC37^{KO}$ 2), $<0.000001$ (*IL6*, WT vs $TTC37^{KO}$ 1), $0.000003$ (*IL6*, WT vs $TTC37^{KO}$ 2). (G) A schematic diagram showing the generation of tamoxifen-inducible *Skiv2l* knockout mouse primary skin-derived fibroblast (MSFb). 4-OHT is a metabolite and the active component of tamoxifen. (H) Western blot analysis of SKIV2L and TTC37 proteins in control and i$Skiv2l^{-/-}$ MSFb cells. (I) RT-qPCR analysis of *Ifnb* and *Il6* mRNA in control and i$Skiv2l^{-/-}$ MSFb cells treated with poly(I:C) (1.0 µg/ml) for 4 h. Data are shown as mean ± SEM ($n = 4$ each genotype). Two-sided Student's $t$ test; ***$P < 0.001$. $P$ values $= 0.000382$ (*Ifnb*), $0.000243$ (*Il6*). (J) Western blot analysis of SKIV2L protein in THES2 patient-derived fibroblasts (pt.) and derivative cells reconstituted with SKIV2L (pt. rescued). (K) RT-qPCR analysis of *IFNB1* and *IL6* mRNA in THES2 patient-derived fibroblasts (pt.) and derivative cells reconstituted with WT SKIV2L (pt. rescued) after poly(I:C) treatment (0.03, 0.1, 0.3 µg/ml) for 4 h. Data are shown as mean ± SEM ($n = 4$ each group). Two-sided Student's $t$ test; **$P < 0.01$, ***$P < 0.001$. $P$ values $= 0.000003$, $0.000314$, $0.000010$ (*IFNB1* from left to right); $0.003988$, $0.000056$, $0.000482$ (*IL6* from left to right). Source data are available online for this figure.

mutant of DExH core failed to reduce elevated IFN response of $SKIV2L^{KO}$ cells after poly(I:C) treatment (Fig. 1D), indicating that the helicase activity of SKIV2L is essential for suppression of dsRNA-induced IFN response. We further stimulated WT and $SKIV2L^{KO}$ cells with high molecular weight (HMW) poly(I:C) or 5' triphosphate hairpin RNA (3p-hpRNA, in vitro transcription of influenza A (H1N1) virus sequence), and also found increased IFN expression in $SKIV2L^{KO}$ cells (Fig. EV1A,B). In addition to IFN mRNA, immunoblot revealed increased protein expression of IFN-stimulated gene (ISG) RSAD2 in $SKIV2L^{KO}$ cells (Fig. EV1C). To test whether $SKIV2L^{KO}$ cells have increased tonic IFN and ISGs (including RIG-I-like receptors) thus priming cells for enhanced RNA sensing, we measured the expression of molecules of RNA sensing pathway at both protein and mRNA levels and found no major difference between WT and $SKIV2L^{KO}$ cells (Fig. EV1D,E). We also measured the expression of a broad panel of ISGs using qPCR and did not observe increased tonic ISG in $SKIV2L^{KO}$ cells (Fig. EV1F). To further corroborate this, we generated $IFNAR1^{KO}SKIV2L^{KO}$ double knockout cells (Fig. EV1G). Compared to $IFNAR1^{KO}$ cells, $IFNAR1^{KO}SKIV2L^{KO}$ cells still exhibited enhanced IFN response after dsRNA stimulation (Fig. EV1H). Collectively, these results rule out the possibility that $SKIV2L^{KO}$ elevates baseline ISG therefore enhancing RNA sensing signaling.

Knockout of *TTC37*, another component of the SKI complex, also significantly enhanced IFN response to poly(I:C) stimulation (Fig. 1E,F). To test whether SKIV2L regulates innate immune response to dsRNA in primary cells, we further generated tamoxifen-inducible *Skiv2l* knockout (i$Skiv2l^{-/-}$) mouse primary skin-derived fibroblasts (MSFb) from $Skiv2l^{fl/fl}UBC-Cre/ERT2$ mice (Fig. 1G,H). We also observed elevated IFN response in *Skiv2l*-deficient MSFb after poly(I:C) stimulation (Fig. 1I). Interestingly, we noticed that loss of SKIV2L led to substantial decrease in TTC37 protein, and vice versa (Fig. 1A,E,H), suggesting both proteins are necessary for maintaining the stability of the SKI complex. Loss-of-function mutations in *SKIV2L* or *TTC37* in humans are associated with a rare inherited disease, Trichohepatoenteric syndrome (THES), characterized by both primary B-cell immunodeficiency

and autoinflammatory features (Yang et al, 2022a; Yang et al, 2022b). We restored SKIV2L expression in *SKIV2L*-deficiency patient-derived fibroblast with stable retroviral transduction (Fig. 1J). Compared with SKIV2L-rescued cells, *SKIV2L*-deficient patient cells showed increase IFN response after dsRNA stimulation (Fig. 1K).

While WT cells tolerated low-dose poly(I:C) stimulation with minimal cell death, we observed that both $SKIV2L^{KO}$ cell lines showed substantial increase in apoptosis, as evidenced by increased Caspase (Casp) 3 and PARP cleavage measured by western blots (Fig. 2A) as well as increased Annexin V staining measured by FACS (Fig. 2B,C). Ectopic expression of wild-type SKIV2L rescued cell death in dsRNA-treated $SKIV2L^{KO}$ cells (Fig. 2D), however E424Q mutant failed to rescue apoptosis of $SKIV2L^{KO}$ cells after dsRNA treatment (Fig. 2E), suggesting that the helicase activity of SKIV2L is essential for limiting dsRNA-induced cell death. Interestingly, the E424Q mutant also exerted a dominant negative effect and enhanced apoptosis in WT cells after dsRNA stimulation (Fig. 2E). Consistently, two independent lines of $TTC37^{KO}$ cells are also more sensitive to dsRNA-induced apoptosis (Fig. 2F). These data suggest that the SKI complex restricts innate immune response to dsRNA.

## SKIV2L restricts the OAS-RNase L pathway

We next investigated the three major cytosolic RNA sensing pathways, RLR-MAVS, OAS-RNase L and PKR-eIF2α, each of which has been implicated in IFN response and/or cell death through distinct mechanisms. We found no major difference in activation of the PKR-eIF2α pathway (measured by phosphorylation of PKR and eIF2α) between WT and $SKIV2L^{KO}$ cells before or after dsRNA stimulation (Figs. 2A and 3A, compare lane 5 to lane 6). This data suggest that SKIV2L does not act on the PKR-eIF2α pathway or directly degrades exogenous dsRNA (which would impact all three RNA sensing pathways).

We next knocked out RNase L or MAVS in $SKIV2L^{KO}$ cells as well as in WT cells to generate a panel of single and double knockouts A549 cells. Ablation of RNase L fully blocked dsRNA-

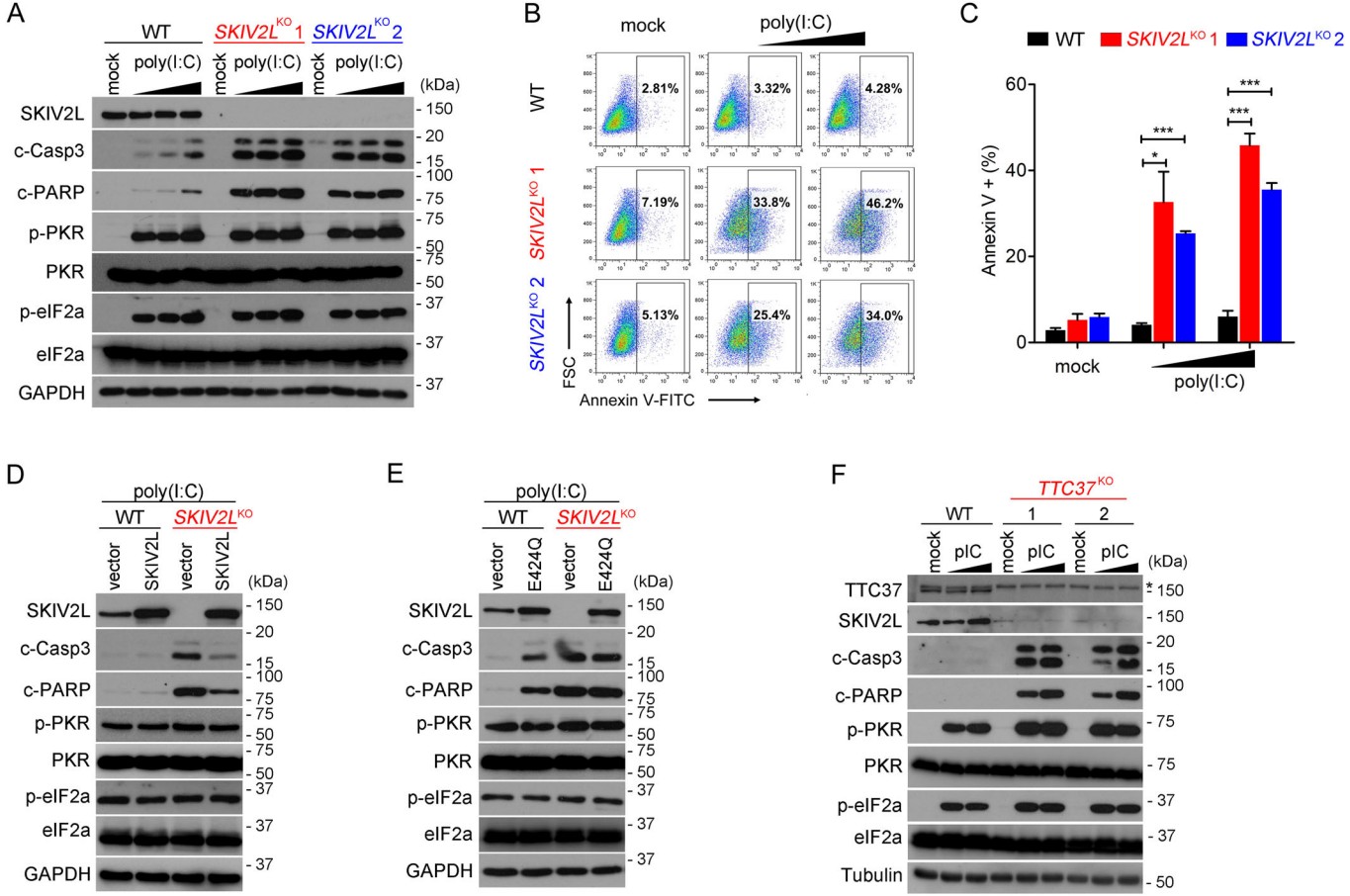

**Figure 2. SKIV2L deficiency enhances dsRNA-induced apoptosis.**

(A) Western blot analysis of apoptosis and PKR-eIF2α pathways in WT and two independent lines of *SKIV2L*[KO] cells after increasing dose of poly(I:C) treatment (0.1, 0.3, 1.0 µg/ml) for 4 h. (B, C) Cell death analysis of WT and two independent lines of *SKIV2L*[KO] cells after poly(I:C) treatment (1.0 µg/ml for 4 h) using Annexin V staining followed by FACS. Representative flow cytometry plot (B) and statistics of three independent experiments (C) were shown. Two-sided Student's *t* test; *P < 0.05, ***P < 0.001. P values = 0.015950, 0.000065 (WT vs *SKIV2L*[KO] 1 from left to right); 0.000201, 0.000757 (WT vs *SKIV2L*[KO] 2 from left to right). (D) Western blot analysis of apoptosis and PKR-eIF2α pathways in WT and *SKIV2L*[KO] cells stably expressing vector or WT SKIV2L after poly(I:C) treatment (1.0 µg/ml for 4 h). (E) Western blot analysis of apoptosis and PKR-eIF2α pathways in WT and *SKIV2L*[KO] cells stably expressing vector or SKIV2L E424Q mutant after poly(I:C) treatment (1.0 µg/ml for 4 h). (F) Western blot analysis of apoptosis and PKR-eIF2α pathways in WT and two independent lines of *TTC37*[KO] cells treated with increasing dose of poly(I:C) treatment (0.3, 1.0 µg/ml) for 4 h. *, a non-specific band. Data are representative of at least three independent experiments. Source data are available online for this figure.

induced apoptosis in *SKIV2L*[KO] cells, suggesting that SKIV2L restricts OAS-RNaseL-mediated apoptosis (Fig. 3A, compare lane 8 to lane 6). There are three catalytically active OAS proteins in humans, and OAS3 is reported to be the major enzyme responsible for RNase L activation (Li et al, 2016). We found that knocking out OAS3 largely reduced dsRNA-induced apoptosis in *SKIV2L*[KO] cells, but with some residual cell death (Fig. 3B). Recent studies, particularly of SARS-CoV-2, have suggested OAS1 to be another contributor of 2–5A production that activates RNase L antiviral pathway (Banday et al, 2022; Soveg et al, 2021; Wickenhagen et al, 2021). We further generated *SKIV2L*[KO]*OAS3*[KO]*OAS1*[KO] triple knockout cells to assess the role of OAS1. Compared to *SKIV2L*[KO]*OAS3*[KO], *SKIV2L*[KO]*OAS3*[KO]*OAS1*[KO] further decreased dsRNA-induced cell death (Fig. EV2A), suggesting a role for OAS1 in RNase L pathway. In contrast to OAS-RNase L pathway, loss of MAVS did not block cell death in *SKIV2L*[KO] cells (Fig. 3C, compare lane 8 to lane 6).

We next measured dsRNA-induced *IFNB1* mRNA expression and grouped single and double knockout cell lines by the presence or absence of SKIV2L. In the *SKIV2L*[WT] group, *RNASEL*[KO] and *OAS3*[KO] showed a slight decrease in poly(I:C)-induced *IFNB1* mRNA expression whereas *MAVS*[KO] completely eliminated *IFNB1* mRNA expression (Fig. 3D,E). This is consistent with the long-standing notion that IFN response to exogenous RNA mostly comes from direct sensing of incoming exogenous RNA ('direct RNA sensing' in Fig. 3D) rather than RNase L-processed RNA ('processed RNA sensing' in Fig. 3D). Surprisingly, in the *SKIV2L*[KO] group, *SKIV2L*[KO] alone produced very high levels of IFN response after poly(I:C) stimulation, which was largely eliminated in *RNASEL*[KO]*SKIV2L*[KO] and *OAS3*[KO]*SKIV2L*[KO] cells (Fig. 3D). In other words, *SKIV2L*[KO] significantly boosted IFN response to exogenous RNA (compared to WT) by expanding the portion of innate immune response from RNase L-processed RNA (Fig. 3F). Of note, *MAVS*[KO] cells still have residual IFN response to poly(I:C), probably through

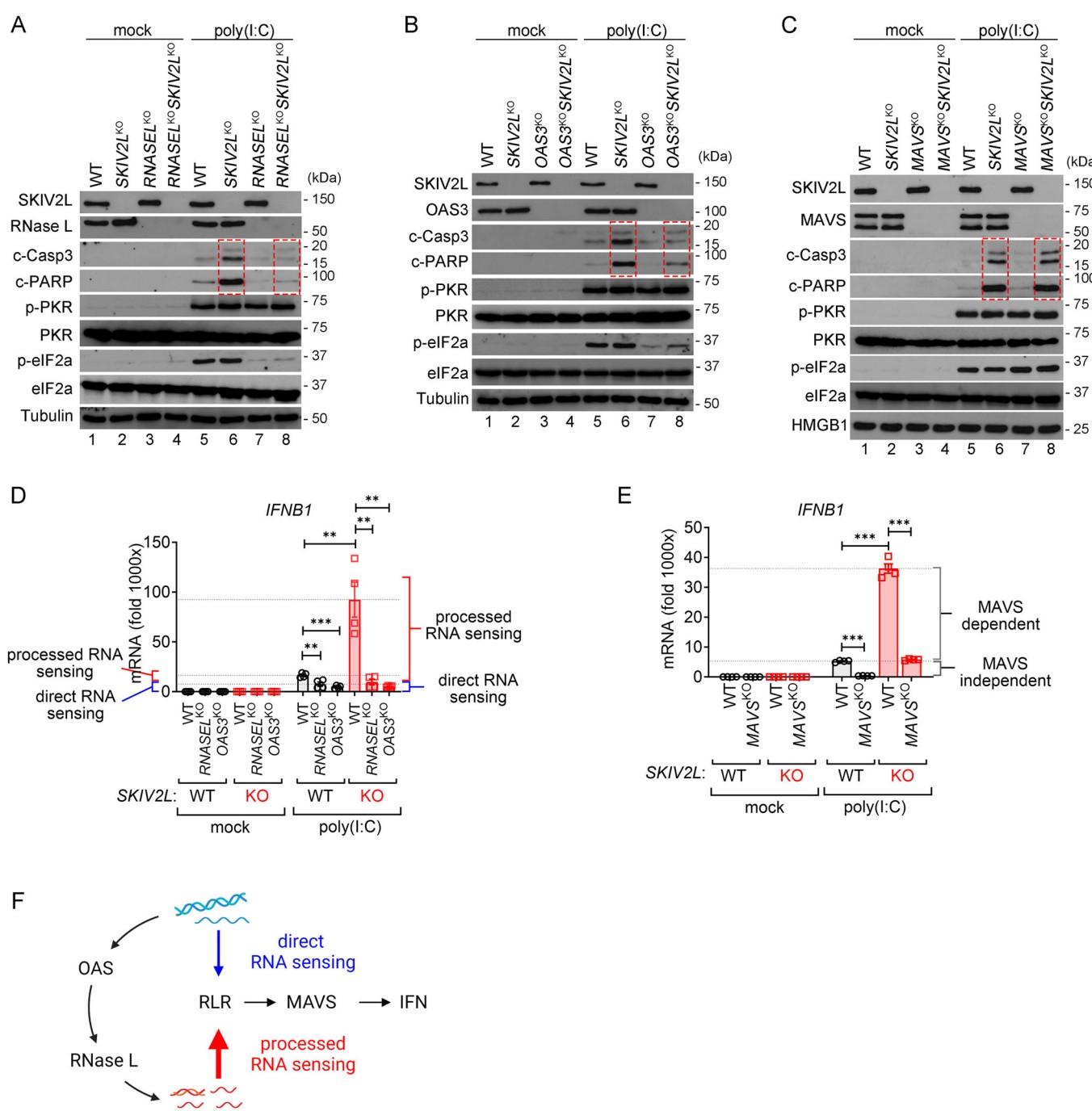

**Figure 3. SKIV2L negatively regulates the OAS-RNase L pathway.**

(A–C) Western blot analysis of apoptosis and PKR-eIF2α pathways in single or double gene knockout cells (as indicated on top) treated with mock or poly(I:C) (1.0 μg/ml) for 4 h. (D, E) RT-qPCR analysis of *IFNB1* mRNA in WT and indicated gene knockout cells after poly(I:C) (1.0 μg/ml) for 4 h treatment. Fold change (1000×) of *IFNB1* mRNA compared to mock-treated WT cells is shown. Data are shown as mean ± SEM of three independent experiments. Two-sided Student's *t* test; **$P < 0.01$, ***$P < 0.001$. (D) *P* values = 0.009380 (WT vs *RNASEL*[KO]), 0.000116 (WT vs *OAS3*[KO]), 0.005046 (WT vs *SKIV2L*[KO]), 0.003540 (*SKIV2L*[KO] vs *SKIV2L*[KO]*RNASEL*[KO]), 0.002554 (*SKIV2L*[KO] vs *SKIV2L*[KO]*OAS3*[KO]). (E) *P* values = 0.000001 (WT vs *SKIV2L*[KO]), <0.000001 (WT vs *MAVS*[KO]), 0.000001 (*SKIV2L*[KO] vs *SKIV2L*[KO]*MAVS*[KO]). (F) Schematic diagram of RLR-MAVS pathway sensing both incoming exogenous RNA and RNase L-processed RNA. Data are representative of at least three independent experiments. Source data are available online for this figure.

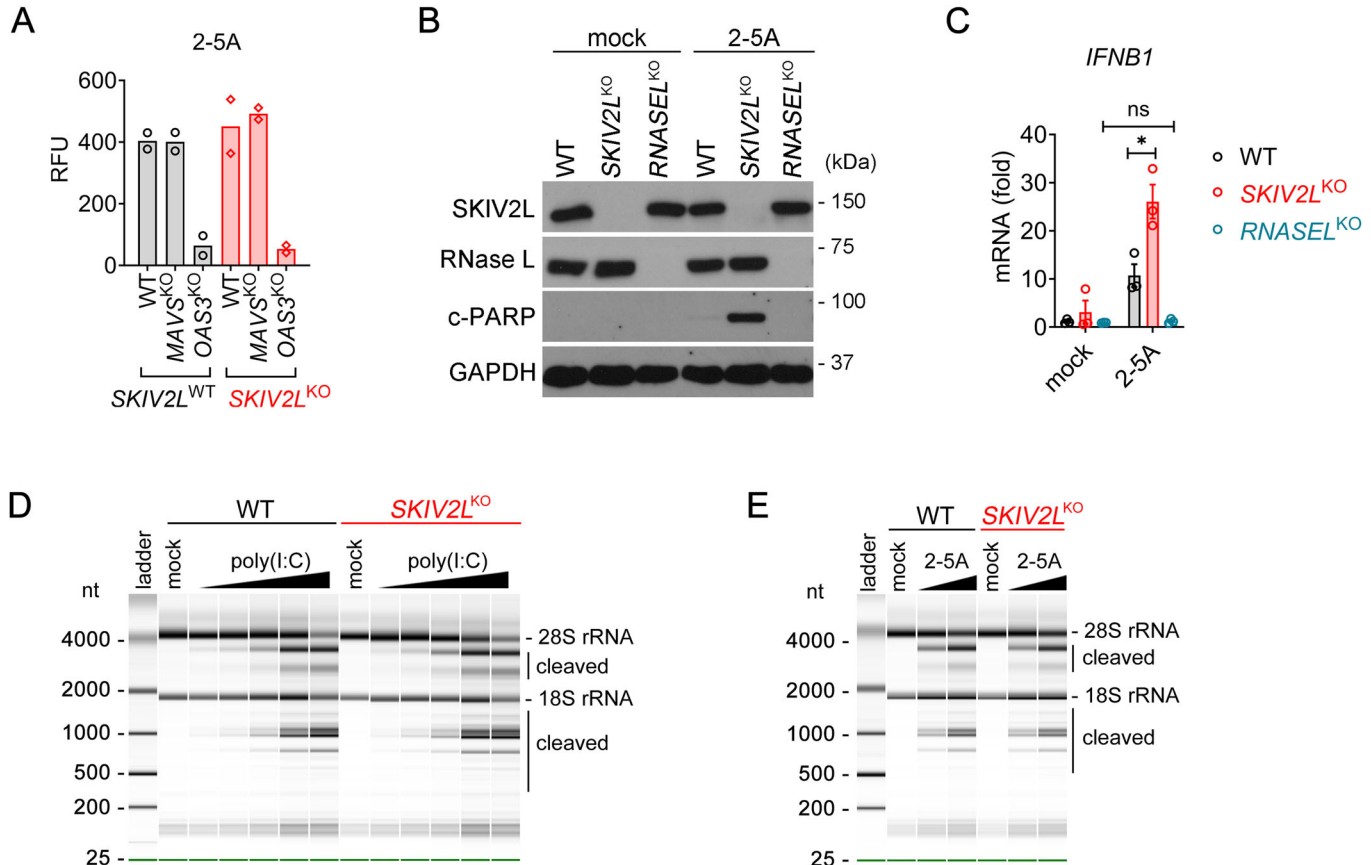

**Figure 4. SKIV2L acts downstream of RNase L.**

(A) Quantification of 2–5A production in WT and knockout cells after poly(I:C) (1 µg/ml) treatment for 4 h. Bars are mean of two independent experiments. (B) Western blot analysis of apoptosis in WT, *SKIV2L*[KO] and *RNASEL*[KO] cells transfected with trimer 2–5A, 2′-5′p3A3 (20 µM) for 4 h. (C) RT-qPCR analysis of *IFNB1* mRNA in WT, *SKIV2L*[KO] and *RNASEL*[KO] cells transfected with 2′-5′p3A3 (20 µM) for 4 h. Data are mean ± SEM of three independent experiments. Two-sided Student's *t* test; *P < 0.05, ns, not significant. P values = 0.022367 (WT vs *SKIV2L*[KO]), 0.292054 (*RNASEL*[KO], mock vs 2–5A). (D, E) rRNA cleavage analysis of WT and *SKIV2L*[KO] cells after increasing dose of poly(I:C) (0.06, 0.12, 0.25, 0.5, 1.0 µg/ml) (D), or 2′-5′p3A3 (10, 20 µM) (E) treatment using Bioanalyzer Nanochip. Data are representative of at least three independent experiments. Source data are available online for this figure.

RLR-MAVS-independent RNA sensing pathway, such as TLR3 (Fig. 3E). We further knocked out RIG-I or MDA5 and found that loss of RIG-I abolished enhanced IFN response in *SKIV2L*[KO] cells (Fig. EV2B,C,D). These data demonstrate that the OAS-RNase L pathway has an underappreciated capability to amplify RNA-induced IFN response, much higher than direct sensing of exogenous RNA (as in WT cells), but most of that capability is restricted by SKIV2L.

## SKIV2L acts downstream of RNase L

We next determined where SKIV2L acts in the OAS-RNase L pathway. OAS senses exogenous RNA, then produces 2–5A, which activates RNase L and subsequent cleavage of cellular and viral RNA. The SKIV2L RNA exosome could act on the RNA species either up- or downstream the OAS-RNase L pathway. We first measured 2–5A production that directly indicates OAS activity and found similar amount of 2–5A in WT and *SKIV2L*[KO] cells after poly(I:C) stimulation (Fig. 4A), suggesting that SKIV2L acts downstream of OAS3. We also performed the complementary experiment to stimulate WT and *SKIV2L*[KO] cells with authentic

trimeric 2–5A (2′,5′p3A3) to directly activate RNase L, and we found that 2′,5′p3A3 stimulated higher apoptosis (Fig. 4B) and IFN response (Fig. 4C) in *SKIV2L*[KO] cells compared to WT, similar to that of poly(I:C) stimulation (Figs. 1 and 2), confirming SKIV2L acts downstream of OAS. Cleavage of ribosomal RNA, a readout of RNase L enzymatic activity after 2–5A binding, was similar in *SKIV2L*[KO] and WT cells after either poly(I:C) (Fig. 4D) or 2–5A stimulation (Fig. 4E), suggesting that SKIV2L does not directly affect RNase L enzymatic activity. Therefore, these data suggest that SKIV2L acts downstream of RNase L likely by removing RNase L-processed RNA products with 2′–3′ cyclic phosphate moiety that would be ideal substrates for the 3′–5′ exoribonuclease activity in the SKIV2L RNA exosome (Zinder et al, 2016).

## SKIV2L restricts antiviral capacity of the OAS-RNase L pathway

To further examine the physiological importance of SKIV2L restriction of the OAS-RNase L pathway during viral infection, we challenged WT, *SKIV2L*[KO], *RNASEL*[KO], *RNASEL*[KO]*SKIV2L*[KO] cells with Sindbis

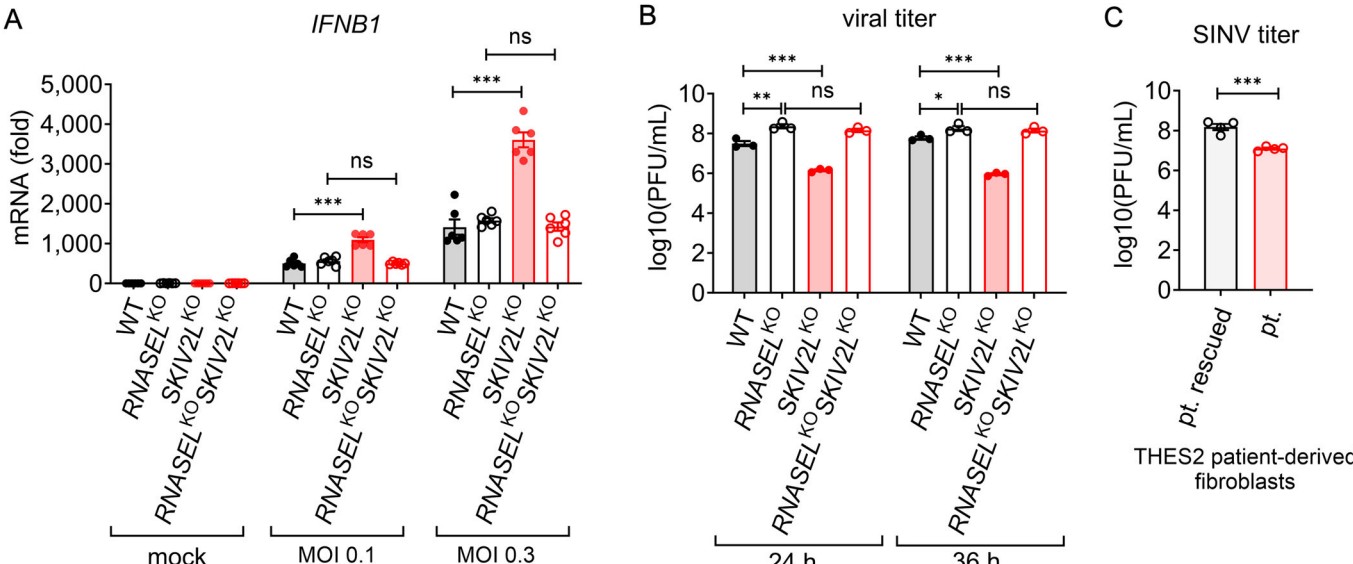

**Figure 5. Loss of SKIV2L enhances RNase L-mediated antiviral activity.**

(A) RT-qPCR analysis of *IFNB1* mRNA in WT, *SKIV2L*KO, *RNASEL*KO and *RNASEL*KO *SKIV2L*KO cells infected with increasing SINV (MOI 0.1, 0.3) for 24 h. Data are mean ± SEM of three independent experiments. Two-sided Student's *t* test; ***P < 0.001; ns, not significant. *P* values = 0.000014 (MOI 0.1, WT vs *SKIV2L*KO), 0.000010 (MOI 0.3, WT vs *SKIV2L*KO), 0.179827 (MOI 0.1, *RNASEL*KO vs *RNASEL*KO*SKIV2L*KO), 0.221373 (MOI 0.3, *RNASEL*KO vs *RNASEL*KO*SKIV2L*KO). (B) Viral production was assessed by plaque assay (PFU, plaque-forming unit) in cells infected with SINV (MOI 0.1) for the indicated time. Data are mean ± SEM of three independent experiments. Two-sided Student's *t* test; *P < 0.05; **P < 0.01; ***P < 0.001; ns, not significant. *P* values = 0.006833 (24 h, WT vs *RNASEL*KO), 0.036438 (36 h, WT vs *RNASEL*KO), 0.000689 (24 h, WT vs *SKIV2L*KO), 0.000044 (36 h, WT vs *SKIV2L*KO), 0.201542 (24 h, *RNASEL*KO vs *RNASEL*KO*SKIV2L*KO), 0.571109 (36 h, *RNASEL*KO vs *RNASEL*KO*SKIV2L*KO). (C) Viral production in THES2 patient-derived fibroblasts (pt.) and derivative cells reconstituted with WT SKIV2L (pt. rescued) after SINV (MOI 0.3) for 24 h. Data are mean ± SEM of four independent experiments. Two-sided Student's *t* test; ***P < 0.001. *P* value = 0.000579. Source data are available online for this figure.

virus (SINV) that is known to activate OAS-RNase L innate immune pathway (Li et al, 2016). Comparing WT to *RNASEL*KO cells, SINV induced a similar amount of IFN response (Fig. 5A), again confirming the long-standing notion of innate sensing of RNase L-processed RNA is largely negligible. *RNASEL*KO cells did show sevenfold higher viral titer at 24 h post-infection than that in WT cells, likely due to cleavage of viral RNA as shown previously (Fig. 5B). In contrast, *SKIV2L*KO cells expressed significantly higher *IFNB* mRNA than WT cells after SINV infection, which was abolished by removing RNase L (*RNASEL*KO*SKIV2L*KO, Fig. 5A). SINV titer was also significantly decreased in *SKIV2L*KO cells by 20- and 66-fold at 24 h and 36 h post-infection, respectively, and this antiviral 'power' was completely lost in *RNASEL*KO*SKIV2L*KO cells (Fig. 5B). We also observed enhanced antiviral activity of *SKIV2L*-deficient patient's fibroblasts compared to SKIV2L-rescued cells after SINV infection (Fig. 5C). These data suggest that SKIV2L limits the antiviral capacity of the OAS-RNase L pathway and removing or inhibiting SKIV2L could potentially release substantial antiviral power from the OAS-RNase L pathway.

## SKIV2L limits OAS1-mediated autoinflammation

Aberrant activation of innate immune RNA sensing pathway, such as gain-of-function (GoF) mutation in RNA sensor OAS1 and loss-of-function (LoF) mutation in RNA editing enzyme ADAR1, has been associated with human inborn errors of immunity (Fig. 6A). *OAS1* GoF variants in human causes a polymorphic autoinflammatory immunodeficiency (Magg et al, 2021). Induction of *OAS1* GoF A76V mutant, but not wild-type *OAS1*, resulted in 2–5A production and

rRNA cleavage in the absence of dsRNA (Figs. 6B,C and EV3A). To test whether SKIV2L regulates pathogenic activation of OAS-RNase L pathway, we induced the expression of OAS1 GoF A76V mutant in WT and *SKIV2L*KO cells (Fig. 6D). We found that *SKIV2L*KO cells were more sensitive to OAS1 GoF mutant-induced cell death (Fig. 6D). *SKIV2L*KO cells also exhibited enhanced IFN response after induction of OAS1 GoF mutant, compared to WT cells (Figs. 6E and EV3B). Deletion of IFNAR1 did not ablate the increase in IFN response in *SKIV2L*KO cells after induction of OAS1 GoF mutant (Fig. EV3C,D), consistent with the findings of dsRNA-stimulated *SKIV2L*KO cells (Fig. EV1G,H). When *RNASEL*, *MAVS* or *DDX58* was deleted, elevated IFN response was reduced in *SKIV2L*KO cells expressing OAS1 GoF mutant (Figs. 6F,G and EV3E). Induction of OAS1 GoF mutant in cells serves as an ideal model to study OAS-RNase L pathway without exogenous dsRNA that can activate other RNA sensing pathway. We found that after induction of OAS1 A76V mutant, cellular RNA isolated from *SKIV2L*KO A549 cells was more immunostimulatory when being transfected to MEFs (Fig. EV3F). *ADAR1* LoF mutations lead to accumulation of endogenous dsRNA that has been reported to activate RLR-IFN and PKR-eIF2a pathways (Chung et al, 2018; Hartner et al, 2009; Liddicoat et al, 2015; Maurano et al, 2021; Rice et al, 2012; Tang et al, 2021). We further generated *SKIV2L*KO*ADAR1*KO cells and found that deletion of *SKIV2L* had no effect on either PKR-eIF2a (Fig. 6H) or IFN response (Fig. 6I) of *ADAR1*KO cells after IFN-β treatment. These results of genetic mutations associated with human autoinflammatory diseases suggest that SKIV2L specifically regulates the OAS-RNase L pathway (Fig. 6J).

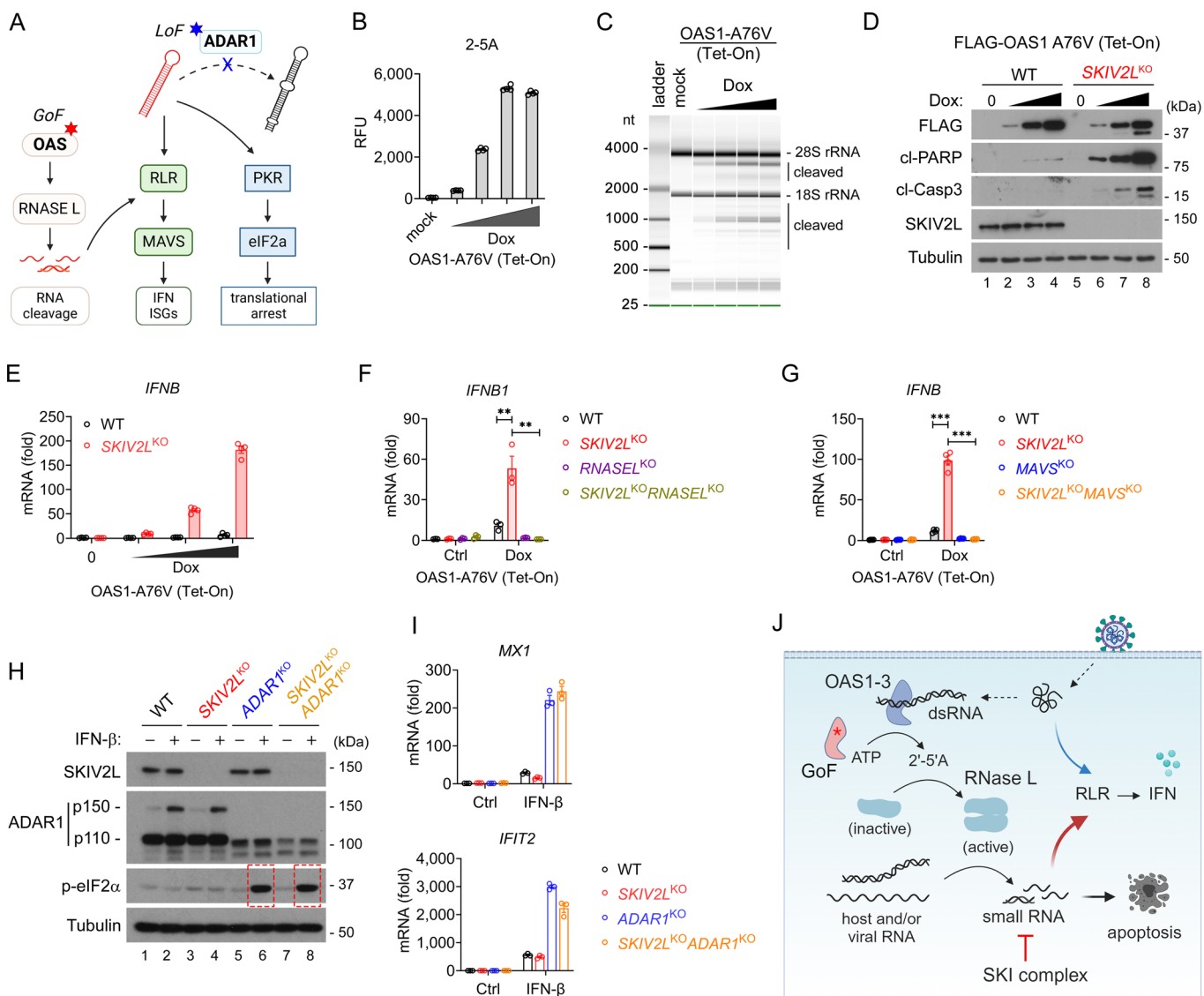

**Figure 6. SKIV2L restricts OAS1-mediated autoinflammation.**

(A) A schematic diagram showing inborn errors of immunity associated with innate immune RNA sensing pathways. (B) Quantification of 2–5 A production in A549 cells after induction of OAS1 A76V mutant with doxycycline (0.25, 0.5, 0.75, 1.0 μg/ml) for 24 h. Data are mean ± SEM of four independent experiments. (C) rRNA cleavage analysis of total RNA in A549 cells after induction of OAS1 A76V mutant with doxycycline (0.25, 0.5, 0.75, 1.0 μg/ml) for 24 h. Data are representative of at least three independent experiments. (D) Western blot analysis of apoptosis markers in WT, *SKIV2L*^KO cells after induction of OAS1 A76V mutant with increasing doses of doxycycline (0.25, 0.5, 1.0 μg/ml) for 24 h. Data are representative of at least three independent experiments. (E) RT-qPCR analysis of *IFNB1* mRNA in WT, *SKIV2L*^KO after induction of OAS1 A76V mutant with doxycycline (0.25, 0.5, 1.0 μg/ml) for 24 h. Data are mean ± SEM of four independent experiments. (F) RT-qPCR analysis of *IFNB1* mRNA in WT, *SKIV2L*^KO, *RNASEL*^KO, *SKIV2L*^KO*RNASEL*^KO cells after induction of OAS1 A76V mutant with doxycycline (1.0 μg/ml) for 24 h. Data are mean ± SEM of three independent experiments. Two-sided Student's *t* test; \*\**P* < 0.01. *P* values = 0.009762 (WT vs *SKIV2L*^KO), 0.004240 (*SKIV2L*^KO vs *SKIV2L*^KO*RNASEL*^KO). (G) RT-qPCR analysis of *IFNB1* mRNA in WT, *SKIV2L*^KO, *MAVS*^KO, *SKIV2L*^KO*MAVS*^KO cells after induction of OAS1 A76V mutant with doxycycline (1.0 μg/ml) for 24 h. Data are mean ± SEM of four independent experiments. Two-sided Student's *t* test; \*\*\**P* < 0.001. *P* values = 0.000005 (WT vs *SKIV2L*^KO), 0.000002 (*SKIV2L*^KO vs *SKIV2L*^KO*MAVS*^KO). (H) Western blot analysis of phosphor-eIF2α in WT, *SKIV2L*^KO, *ADAR1*^KO, *SKIV2L*^KO*ADAR1*^KO cells after IFN-β treatment (20 ng/ml) for 24 h. (I) RT-qPCR analysis of ISGs in WT, *SKIV2L*^KO, *ADAR1*^KO, *SKIV2L*^KO*ADAR1*^KO cells after IFN-β treatment (20 ng/ml) for 24 h. Data are mean ± SEM of three independent experiments. (J) A schematic diagram summarizing SKIV2L regulation of the OAS-RNase L pathway. Source data are available online for this figure.

# Discussion

A long-standing conundrum of the OAS-RNase L pathway is its powerful enzymatic amplification of RIG-I-compatible 3'RNA ends (through RNase L cleavage of cellular and viral RNA) and yet its modest contribution to the overall IFN production during infection (Malathi et al, 2007). We show here that the human SKI complex, a cofactor complex that recruits RNA substrates for degradation by the cytoplasmic RNA exosome, plays a crucial role in limiting the antiviral activities of the OAS-RNase L pathway. This evolutionarily conserved RNA degradation machinery, normally involved in nonsense mediated decay of cellular RNA, now also acts as an unfortunate barrier to antiviral immunity. When this barrier is removed (e.g. *SKIV2L*^KO cells), the OAS-RNase L pathway becomes

fully unleashed and dominates the IFN response to exogenous RNA challenge (via RIG-I sensing of RNase L-processed RNA) and far exceeds that from RIG-I direct sensing of incoming RNA. On the other hand, the SKI complex may serve as a gatekeeper to prevent overactivation of innate immunity and inadvertent immune pathologies (e.g. cell death) in both viral infections and autoinflammation.

The mechanism described here has important parallels to a previous study (Eckard et al, 2014). In cells under endoplasmic reticulum (ER) stress, *SKIV2L* knockdown enhances RIG-I-mediated IFN response due to the accumulation of ribonuclease-IRE1-processed cellular RNA (Eckard et al, 2014). IRE1 and RNase L are two highly homologous ribonucleases, both activated by an upstream signal (unfolded proteins activate IRE1, 2–5A activate RNase L), both dimerize and then cleave substrate RNAs, and both are metal-ion-independent endoribonucleases that produces 3′ cyclic phosphate RNA ends that activate RIG-I (Jung et al, 2020; Shigematsu et al, 2018). The nuclease domain of RNase L also shares structural similarity to IRE1 (Dong et al, 2001). The one difference is that IRE1 is membrane anchored in the ER and RNase L is ubiquitously present in the cytosol, which makes sense for the upstream stimuli they each detect. Therefore, we propose that, during ER stress, the cytoplasmic SKIV2L RNA exosome degrades IRE1-processed RNA to prevent innate immune activation (as a protective mechanism for the host); during RNA virus infection, it similarly degrades RNase L-processed RNA thus (unfortunately) limiting antiviral immunity.

The nature of RNA substrates cleavage by RNase L and then rapidly degraded by SKIV2L RNA exosome remain unclear. We showed here that the RNA substrates accumulate in SKIV2L-deficient cells are immunogenic and activates RLR. One previous study in *Drosophila* cells has shown that the nuclear RNA exosome can also target viral RNA for degradation (Molleston et al, 2016). However, it remains unclear whether the mammalian cytoplasmic SKIV2L RNA exosome has a similar capacity to degrade viral RNA directly without prior RNase L cleavage. Extensive structural studies have elucidated the mechanism of RNA recognition by RLRs (Cadena and Hur, 2019; Chen and Hur, 2022; Rehwinkel and Gack, 2020). RIG-I filament assembly and oligomerization require dsRNA with a 5′-triphosphate (5′ppp) or 5′-diphosphate group, which is structurally characterized in 5′pppRNA-bound RIG-I multimer (Goubau et al, 2014; Myong et al, 2009; Peisley et al, 2013). 3′ cyclic phosphate RNA ends produced by metal-ion-independent endoribonucleases can also activate RIG-I (Jung et al, 2020; Shigematsu et al, 2018). Further, rearrangement of the actin cytoskeleton can facilitate RLR activation by promoting the dephosphorylation of RIG-I and MDA5 (Acharya et al, 2022).

Therefore, a major possibility is that RNA virus activates OAS-RNase L, which cleavages both cellular and viral RNA and generates 3′ cyclic phosphate RNA ends that are either degraded by SKIV2L RNA exosome or sensed by RLR. A minor possibility is that SKIV2L RNA exosome directly degrades viral RNA and prevents them from triggering RLR.

We believe that SKIV2L-mediated regulation of RLR pathway is related to its function within SKI complex of RNA exosome, although our data does not exclue the possibility that SKIV2L may also have activity outside of the RNA exosome. We showed that the *SKIV2L*KO and *TTC37*KO cells phenocopied each other in all assays tested in our study. In both human and mouse cells, knockout of either SKIV2L or TTC37 of SKI complex destablized the whole complex and resulted in decrease protein level of the other component. While cryo-EM structural studies have elegantly elucidated the mechanism of RNA degradation by SKI complex and cytoplasmic RNA exosome (Halbach et al, 2013; Schmidt et al, 2016), it is intriguing to explore the function and potential RNA substrates of "free" SKIV2L outside of SKI complex.

The discovery of this cell-intrinsic barrier to antiviral immunity immediately opens up opportunities for therapy. Interestingly, a recent yeast suppressor screen identified the human SKI complex as a promising broad-spectrum antiviral drug target (Weston et al, 2020). Inhibition of SKIV2L RNA helicase activity by chemical compounds impaired the replication of influenza, filoviruses as well as several human coronaviruses (SARS-CoV, MERS-CoV and SARS-CoV-2) (Weston et al, 2020). The underlying mechanism is not yet defined, although based on our data presented here, we predict that the OAS-RNase L pathway may underlie the antiviral activity of SKIV2L inhibitors. Together, our results not only reveal the enormous hidden power of the OAS-RNase L signaling pathway, but also a targetable host barrier protein SKIV2L for antiviral therapy against many RNA viruses that cause severe diseases in humans.

# Methods

## Methods and protocols

### Cell culture
A549 cells, Vero, BHK-21 and HEK293T cells were maintained in DMED supplemented with 10% FBS. RNase L knockout and OAS3 knockout A549 cells were generated as described previously and kindly provided by Dr. Susan R. Weiss (University of Pennsylvania)

**Reagents and tools table**

| Reagent/resource | Reference or source | Identifier or catalog number |
| --- | --- | --- |
| **Experimental models** | | |
| A549 cells (*H. sapiens*) | ATCC | CCL-185 |
| 293T (*H. sapiens*) | ATCC | CRL-3216 |
| BHK-21 (*Mesocricetus auratus*) | ATCC | CCL-10 |
| Vero cells (*Chlorocebus sabaeus*) | ATCC | CCL-81 |

| Reagent/resource | Reference or source | Identifier or catalog number |
|---|---|---|
| SKIV2L-deficient THES2 patient-derived fibroblasts (*H. sapiens*) | (Yang et al, 2022a) | N/A |
| *Skiv2l*<sup>fl/fl</sup> (*M. musculus*) | (Yang et al, 2022a) | N/A |
| *Skiv2l*<sup>fl/fl</sup>*UBC-Cre/ERT2* (*M. musculus*) | (Yang et al, 2022a) | N/A |
| Sindbis virus | (Orvedahl et al, 2010) | N/A |
| Sendai virus | (Hasan et al, 2013) | N/A |
| Encephalomyocarditis virus | (Aguilera et al, 2019) | N/A |
| **Recombinant DNA** | | |
| LentiCRISPRv2 | Addgene | 52961 |
| pMRX-SKIV2L | This study | N/A |
| pMRX-SKIV2L-E424Q | This study | N/A |
| pEasiLV-OAS1 | This study | N/A |
| pEasiLV-OAS1-A76V | This study | N/A |
| **Antibodies** | | |
| SKIV2L Rabbit Polyclonal Antibody | Proteintech | 11462-1-AP |
| TTC37 Rabbit Polyclonal Antibody | Proteintech | 24594-1-AP |
| Monoclonal Anti-α-Tubulin antibody (clone B-5-1-2) | Sigma | T5168 |
| Anti-HMGB1 antibody | Abcam | ab18256 |
| GAPDH (14C10) Rabbit mAb | Cell Signaling | 2118 |
| Vinculin Antibody | Cell Signaling | 4650 |
| OAS3 Rabbit Polyclonal Antibody | Proteintech | 21915-1-AP |
| OAS1 Recombinant antibody | Proteintech | 82883-1-RR |
| RNase L (D4B4J) Rabbit mAb | Cell Signaling | 27281 |
| RIG-I (D14G6) Rabbit mAb | Cell Signaling | 3743 |
| MDA-5 (D74E4) Rabbit mAb | Cell Signaling | 5321 |
| MAVS (D5A9E) Rabbit mAb | Cell Signaling | 24930 |
| TBK1/NAK (D1B4) Rabbit mAb | Cell Signaling | 3504 |
| IRF-3 (D6I4C) XP® Rabbit mAb | Cell Signaling | 11904 |
| Viperin (D5T2X) Rabbit mAb mAb | Cell Signaling | 13996 |
| Recombinant Anti-Interferon alpha/beta receptor 1 antibody [EPR6244] | Abcam | ab124764 |
| Cleaved Caspase-3 (Asp175) (5A1E) Rabbit mAb | Cell Signaling | 9664 |
| Cleaved PARP (Asp214) (D64E10) Rabbit mAb | Cell Signaling | 5625 |
| PKR (phospho T446) antibody [E120] | Abcam | ab32036 |
| PKR (D7F7) Rabbit mAb | Cell Signaling | 12297 |
| Phospho-eIF2α (Ser51) (119A11) Rabbit mAb | Cell Signaling | 3597 |

| Reagent/resource | Reference or source | Identifier or catalog number |
|---|---|---|
| eIF2α (D7D3) XP® Rabbit mAb | Cell Signaling | 5324 |
| ADAR1 mouse monoclonal antibody (15.8.6) | Santa Cruz | sc-73408 |
| Goat anti-rabbit IgG (H/L):HRP | Bio-Rad | 5196-2504 |
| Goat anti-mouse IgG (H/L):HRP | Bio-Rad | 5178-2504 |
| **Oligonucleotides and other sequence-based reagents** | | |
| Human *SKIV2L* gRNA1 | This study | 5′-GACGGATCCCTGGTCTCTTT-3′ |
| Human *SKIV2L* gRNA2 | This study | 5′-CTTTGGGCCTGTAGGTCGGA-3′ |
| Human *TTC37* gRNA1 | This study | 5′-TGGTGTTTACCAAAAGCTCC-3' |
| Human *TTC37* gRNA2 | This study | 5′-TGATGTCTGCAAGAAACTTG-3′. |
| Human *MAVS* gRNA | This study | 5′-CTGTGAGCTAGTTGATCTCG-3′ |
| Human *ADAR1* gRNA | This study | 5′- TCTGTCAAATGCCATATGGG-3′ |
| *OAS1*-crRNA1 | This study | /AltR1/rArGrUrArCrGrArArGrCrUrGrArGrCrGrCrArCrGrGrUrUrUrArGrArGrCrUrArUrGrCrU/ AltR2/ |
| *OAS1*-crRNA2 | This study | /AltR1/rGrCrUrCrCrArArGrCrArUrArGrArCrArCrGrUrCrGrUrUrUrUrArGrArGrCrUrArUrGrCrU/ AltR2/ |
| *IFNAR1*-crRNA1 | This study | /AltR1/rGrCrGrGrCrUrGrCrGrGrArCrArArArCrArCrCrCrArGrUrUrUrUrArGrArGrCrUrArUrGrCrU/ AltR2 |
| *IFNAR1*-crRNA2 | This study | /AltR1/rArArGrCrArGrCrArCrUrArCrUrUrArCrGrUrCrArGrUrUrUrUrArGrArGrCrUrArUrGrCrU/ AltR2/ |
| *IFIH1*-crRNA1 | This study | /AltR1/rUrCrArUrGrArGrCrGrUrUrCrUrCrArArArCrGrArGrUrUrUrUrArGrArGrCrUrArUrGrCrU/ AltR2/ |
| *IFIH1*-crRNA2 | This study | /AltR1/rUrUrGrGrArCrUrCrGrGrGrArArUrUrCrGrUrGrGrGrUrUrUrUrArGrArGrCrUrArUrGrCrU/ AltR2/; |
| *DDX58*-crRNA1 | This study | /AltR1/rGrGrArUrUrArUrArUrCrCrGrGrArArGrArCrCrCrGrUrUrUrUrArGrArGrCrUrArUrGrCrU/ AltR2/ |
| *DDX58*-crRNA2 | This study | /AltR1/rGrArUrCrArGrArArArUrGrArUrArUrCrGrGrUrUrGrUrUrUrArGrArGrCrUrArUrGrCrU/ AltR2/ |
| Alt-R® CRISPR-Cas9 tracrRNA | Integrated DNA Technologies | 1072532 |
| PCR primers | (Tu et al, 2022; Wu et al, 2020; Yang et al, 2018) | |
| **Chemicals, enzymes and other reagents** | | |
| TRI reagent | Sigma | T9424 |
| iScript™ cDNA Synthesis Kit | Bio-Rad | 1708890 |
| iTaq™ Universal SYBR® Green Supermix | Bio-Rad | 1725122 |
| Dulbecco's Modified Eagle's Medium - high glucose | Sigma | D5796 |
| Fetal bovine serum | Sigma | F2442 |
| Dulbecco's Phosphate Buffered Saline | Sigma | D8537 |
| Gibco™ Puromycin Dihydrochloride | Thermo Fisher | A1113803 |
| Blasticidin S, Hydrochloride | Sigma | 15205 |
| Poly(I:C) (LMW) | Invivogen | tlrl-picw |
| Poly(I:C) (HMW) | Invivogen | tlrl-pic |
| 2′-5′p3A3 | This study | N/A |

| Reagent/resource | Reference or source | Identifier or catalog number |
|---|---|---|
| Doxycycline Hyclate | Sigma | D5207 |
| Recombinant Human IFN-β | PeproTech | 300-02BC |
| Alt-R S.p. Cas9 Nuclease V3 | Integrated DNA Technologies | 1081058 |
| Lipofectamine™ 2000 Transfection Reagent | Thermo Fisher | 11668500 |
| Lipofectamine RNAiMAX reagent | Thermo Fisher | 13778100 |
| FITC-Annexin V Apoptosis Detection Kit | BioLegend | 640914 |
| SuperSignal™ West Pico PLUS Chemiluminescent Substrate | Thermo Fisher | 34580 |
| miRNeasy Tissue/Cells Advanced Kits | QIAGEN | 217684 |
| **Software** | | |
| FlowJo 10.6 | FlowJo, LLC | |
| GraphPad Prism 10.1.1 | GraphPad Software | |
| BioTek Gen5 Software | BioTek | |
| **Other** | | |
| BD FACSCalibur™ Flow Cytometer | BD Biosciences | |
| ChemiDoc Imaging Systems | Bio-Rad | |
| CFX96™ Real-Time PCR Detection System | Bio-Rad | |
| 2100 Bioanalyzer system | Agilent | |

(Li et al, 2016). SKIV2L-deficient THES2 patient-derived fibroblasts were described previously (Yang et al, 2022a). Mouse primary skin-derived fibroblasts (MSFb) were generated from *Skiv2l*$^{fl/fl}$ and *Skiv2l*$^{fl/fl}$*UBC-Cre/ERT2* mice (Yang et al, 2022a) following the protocol described previously (Khan and Gasser, 2016). All mice were housed in pathogen-free barrier facilities at UT Southwestern Medical Center. The animal protocol was approved by the Institutional Animal Care and Use Committee at UT Southwestern Medical Center (APN 2017-101968). Ex vivo deletion of *Skiv2l* in culture cells was described previously (Yang et al, 2022a). All cells used in the study were tested negative for mycoplasma contamination.

### CRISPR/Cas9-mediated gene editing

Knockout cell lines were generated using CRISPR/Cas9-mediated gene editing either through lentivrial transduction or transient cationic lipid delivery of CRISPR/Cas9 ribonucleoprotein (RNP) complex. For CRISPR/Cas9 lentivrial transduction, sgRNAs were designed and cloned into LentiCRISPRv2 vector. The packaging of pseudo lentiviruses carrying sgRNA and Cas9 was described previously (Yang et al, 2018). A549 cells were transduced with lentiviruses, followed by puromycin (2 μg/ml) selection for several days.

For transient CRIPSR/Cas9 RNP transfection, the following crRNAs were purchased from. Each crRNA was annealed with Alt-R® CRISPR-Cas9 tracrRNA to form crRNA: tracrRNA duplex, then

incubate with to assemble RNP complex. RNP complex was transfected into cells using Lipofectamine RNAiMAX reagent. Single cell-derived clones were confirmed with western blot for knockout of gene of interest.

### Retrovirusl and lentivirus preparation and transduction

SKIV2L were cloned into retroviral pMRX-ires-bsr vector (a kind gift from S. Akira) using *EcoR* I and *Not* I sites. SKIV2L E424Q mutant was generated by site-directed mutagenesis (Agilent Technologies). Synonymous mutations were introduced in *SKIV2L* sgRNA targeting sequences and protospacer adjacent motif (PAM). Human OAS1 A76V mutant was cloned into pEsiLV lentiviral vector as described previously (Wu et al, 2019). Retroviruses and lentiviruses were packaged in HEK293T cells following standard protocol. Retroviruses were used for transduction followed by selection with blasticidine (15 μg/ml) for 7 days. pEsiLV lentiviruses were used for transduction followed by single-cell clone selection. Cells transduced with human OAS1 A76V were verified with western blot and flow cytometry after doxycycline induction.

### Transfections with poly(I:C) or 2–5A

Transfections of poly(I:C) and 2-5A were performed with Lipofectamine 2000 per manufacturer's instruction. Briefly, poly(I:C) or 2–5A diluted in Opti-MEM was mixed with diluted Lipofectamine 2000 reagent to form complex, then added to

sub-confluent cells. Lipofectamine alone was used as mock control. Cells were collected at indicated time points for RT-qPCR or western blot analysis.

### RNA isolation and RT-qPCR

Total RNA was isolated from cultured cells using TRI reagent (Sigma) per manufacturer's instruction, and cDNA were synthesized with iScript cDNA Synthesis Kit (Bio-Rad). iTaq Universal SYBR Green Supermix (Bio-Rad) was used to quantify mRNA expression with CFX96™ Real-Time PCR Detection System.

### Small RNA isolation and transfection

The isolation of small RNA was performed as described previously (Malathi et al, 2007). Briefly, OAS1 A76V mutant was induced in WT and SKIV2LKO A549 for 24 h. Small RNAs were isolated using miRNeasy Tissue/Cells Advanced Kits. Small RNAs were transfected into MEFs using Lipofectamine 2000 and induction of mouse *Ifnb* was analyzed using RT-qPCR.

### Western blotting

Western blots were performed as described previously (Yang et al, 2018). Briefly, cell lysate was quantified using BCA and equal amounts of proteins were separated on SDS-polyacrylamide gel and transferred to nitrocellulose membrane. Membranes were blocked with 5% non-fat milk in 1X TBS-T and incubated with diluted primary antibodies at 4 °C overnight per manufacturers' instructions. Membranes were incubated with HRP-conjugated secondary antibody (Bio-Rad) diluted for 1 h at room temperature. Super-Signal West Pico Chemiluminescent Substrate (Thermo Scientific) was used to develop the blots on X-ray film or using ChemiDoc™ Imaging System (Bio-Rad).

### Annexin V apoptosis assay

Apoptosis was measured using FITC-Annexin V Apoptosis Detection Kit. Briefly, cells were transfected with poly(I:C) at indicated concentration for 4 h, and then collected and washed twice with cold PBS. Cells were resuspended in Annexin V Binding Buffer and stained with FITC-Annexin V for 15 min at room temperature in the dark. Stained cells were analyzed by flow cytometry (BD FACSCalibur) and data were analyzed using FlowJo software.

### Quantification of 2-5A

Intracellular 2-5A was quantified by an indirect RNase L-based FRET assays as described before (Thakur et al, 2005). Briefly, poly(I:C)-treated cells were washed with PBS, and lysed in preheated (95 °C) Nonidet P-40 lysis buffer (50 mM Tris-HCl, pH 7.2, 0.15 M NaCl, 1% Nonidet P-40, 200 mM sodium orthovanadate, 2 mM EDTA, 5 mM $MgCl_2$, 5 mM DTT) and heated to 95 °C for another 7 min. The cleared supernatants collected after centrifugation at $14{,}000 \times g$ for 10 min. Levels of 2-5A were determined by RNase L-based FRET assays with recombinant human RNase L and synthetic fluorophore-labeled oligoribonucleotide as substrate.

### rRNA cleavage assay

Total RNA was isolated using TRI reagent (Sigma) per manufacturer's instruction. Equal amount of total RNA was then resolved on RNA nanochips using an Agilent 2100 BioAnalyzer.

### Plaque assays

SINV was diluted serially in DMEM and 250 μL of diluted viruses were added to confluent Vero cell monolayers in six-well plates. The plates were incubated for 1 h at 37 °C with rocking at 15-min intervals. Then the cells were overlaid with 3 mL warm DMEM containing 2% FBS and 1% Agar. After 36-48 h incubation at 37 °C with 5% $CO_2$, cells were fixed with 4% formaldehyde and plaques were visualized using 0.1% Crystal violet staining.

### Statistical analysis

For statistical analyses, most of the experiments were repeated three or more times as indicated in each figure legend. The sample size was not pre-determined in this study. No data were excluded from the analyses. Investigators were not blinded during data collection. Graphpad Prism was used for statistical analysis. Statistical tests performed were indicated in figure legend. Numerical data were shown as mean ± SEM. $P$ values of less than 0.05 were considered statistically significant.

## Data availability

This study includes no data deposited in external repositories.

The source data of this paper are collected in the following database record: biostudies:S-SCDT-10_1038-S44318-024-00187-1.

## Peer review information

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

## Acknowledgements

We thank Dr. Susan Weiss (U Penn) for providing *RNASEL*[KO] and *OAS3*[KO] A549 cells, members of the Yan lab for helpful discussion. We thank Drs. Julie Pfeiffer and Carolyn Sturge (UT Southwestern Medical Center) for assistance with EMCV infection. This work was supported by the NIH (AI153576 to NY, AI135922 to RHS).

## Author contributions

**Kun Yang**: Conceptualization; Data curation; Formal analysis; Validation; Investigation; Visualization; Methodology; Writing—original draft; Project administration; Writing—review and editing. **Beihua Dong**: Formal analysis; Investigation. **Abhishek Asthana**: Formal analysis; Investigation. **Robert H Silverman**: Conceptualization; Resources; Formal analysis; Methodology; Writing—original draft. **Nan Yan**: Conceptualization; Resources; Data curation; Formal analysis; Supervision; Funding acquisition; Investigation; Visualization; Writing—original draft; Project administration; Writing—review and editing.

Source data underlying figure panels in this paper may have individual authorship assigned. Where available, figure panel/source data authorship is listed in the following database record: biostudies:S-SCDT-10_1038-S44318-024-00187-1.

## Disclosure and competing interests statement

The authors declare no competing interests.

# Expanded View Figures

**Figure EV1.** *SKIV2L* **deficiency enhances dsRNA-induced type I IFN response.**

(A) RT-qPCR analysis of *IFNB1* mRNA in WT and two independent lines of *SKIV2L*$^{KO}$ cells treated with high molecular weight (HMW) or low molecular weight (LMW) poly(I:C) (1.0 μg/ml) for 4 h. Data are shown as mean ± SEM of three independent experiments. Two-sided Student's *t* test; \*\*\**P* < 0.001. *P* values = 0.000008 (HMW, WT vs *SKIV2L*$^{KO}$ 1), 0.000123 (LMW, WT vs *SKIV2L*$^{KO}$ 1), 0.000965 (HMW, WT vs *SKIV2L*$^{KO}$ 2), 0.000293 (LMW, WT vs *SKIV2L*$^{KO}$ 2). (B) RT-qPCR analysis of *IFNB1* mRNA in WT and *SKIV2L*$^{KO}$ cells treated with 3p-hpRNA at indicated concentration for 4 h. Data are shown as mean ± SEM of four independent experiments. Two-sided Student's *t* test; \*\**P* < 0.01, \*\*\**P* < 0.001. *P* values = 0.000007 (150 ng/ml), 0.000038 (500 ng/ml). (C) Western blot analysis of RSAD2 proteins in WT and two independent lines of *SKIV2L*$^{KO}$ A549 cells after poly(I:C) (0.1 μg/ml) treatment for 8 h. (D) Western blot analysis of proteins of RNA sensing pathway in WT and two independent lines of *SKIV2L*$^{KO}$ A549 cells. (E) RT-qPCR analysis of genes of RNA sensing pathway in WT and two independent lines of *SKIV2L*$^{KO}$ A549 cells. Data are shown as mean ± SEM of three independent experiments. (F) RT-qPCR analysis of ISG expression in WT and two independent lines of *SKIV2L*$^{KO}$ A549 cells. Data are shown as mean ± SEM of three independent experiments. (G) Western blot analysis of IFNAR1 and SKIV2L proteins in WT and indicated knockout cells. (H) RT-qPCR analysis of *IFNB1* mRNA in WT and indicated knockout cells after poly(I:C) (1.0 μg/ml) treatment for 4 h or SINV infection (MOI 0.3) for 24 h. Data are shown as mean ± SEM of four independent experiments. Two-sided Student's *t* test; \*\*\**P* < 0.001. *P* values = 0.000003 (poly(I:C), *IFNAR1*$^{KO}$ vs *IFNAR1*$^{KO}$*SKIV2L*$^{KO}$ 1), 0.000283 (poly(I:C), *IFNAR1*$^{KO}$ vs *IFNAR1*$^{KO}$*SKIV2L*$^{KO}$ 2), 0.000366 (SINV, *IFNAR1*$^{KO}$ vs *IFNAR1*$^{KO}$*SKIV2L*$^{KO}$ 1), 0.000002 (SINV, *IFNAR1*$^{KO}$ vs *IFNAR1*$^{KO}$*SKIV2L*$^{KO}$ 2). (I) RT-qPCR analysis of *IFNB1* and ISGs mRNA in WT and two independent lines of *SKIV2L*$^{KO}$ A549 cells after recombinant IFN-b treatment (20 ng/ml for 24 h). Data are shown as mean ± SEM of three independent experiments. Source data are available online for this figure.

▶

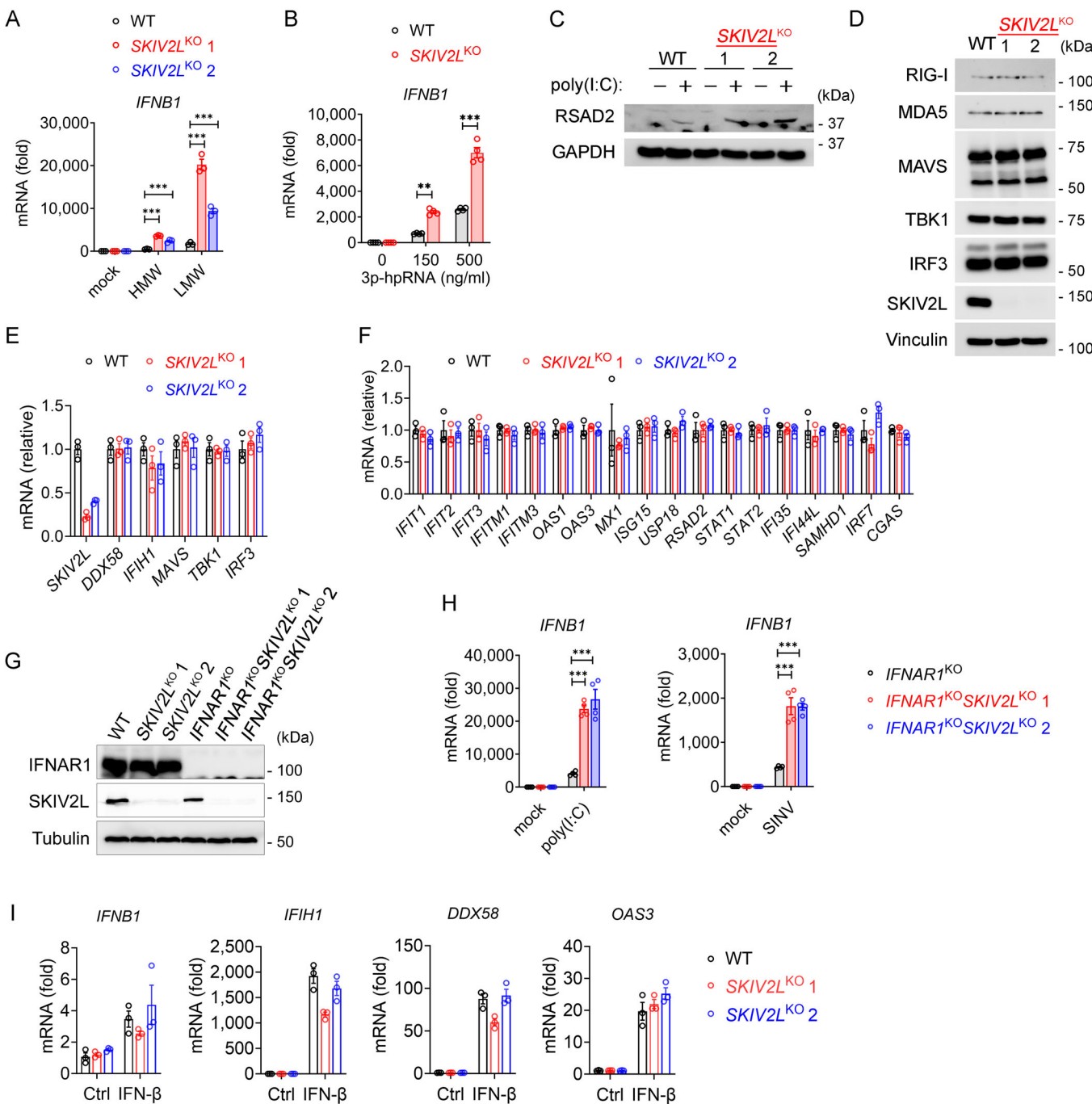

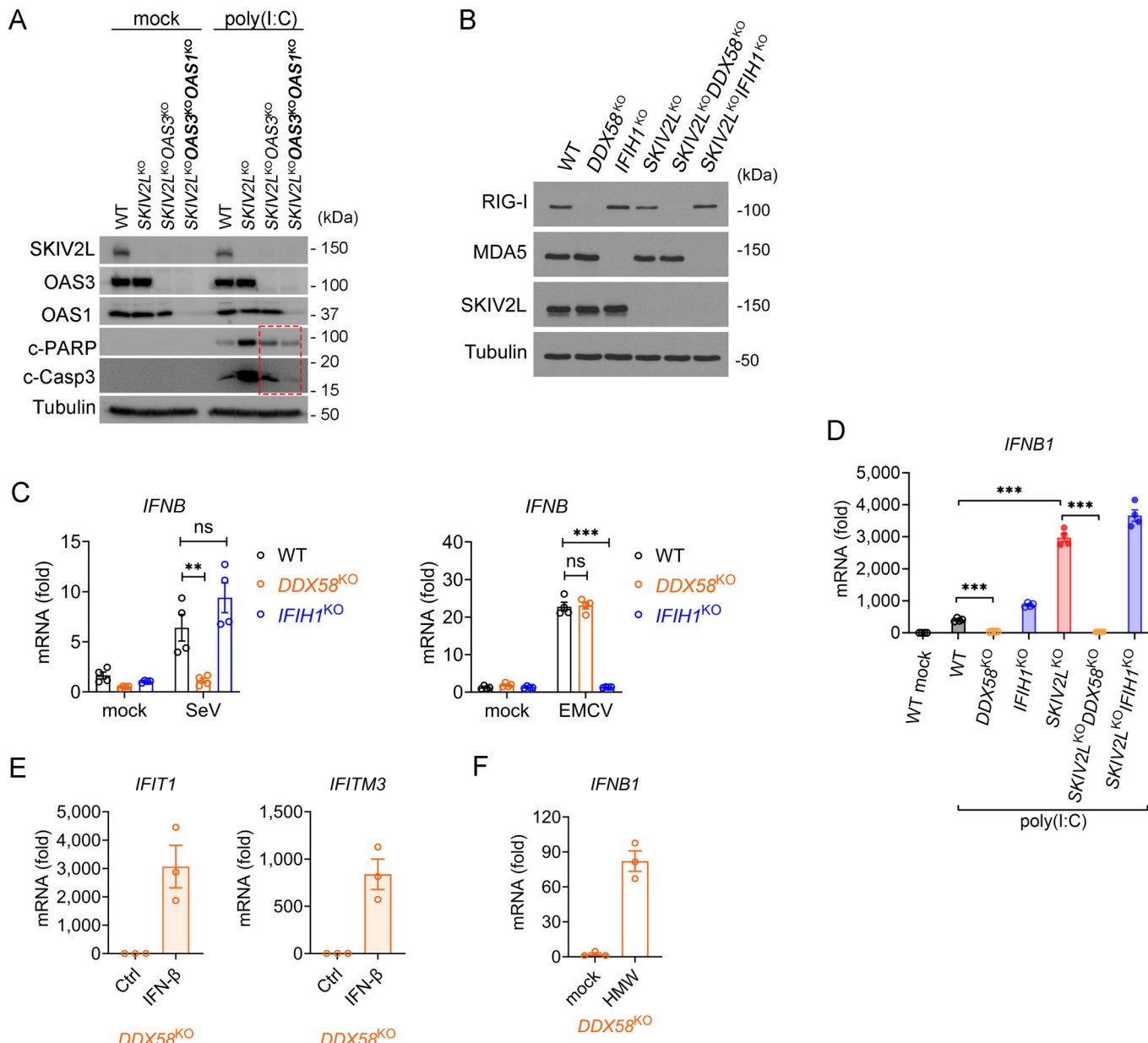

**Figure EV2. SKIV2L negatively regulates the OAS-RNase L pathway.**

(A) Western blot analysis of apoptosis in WT, *SKIV2L*KO, *SKIV2L*KO*OAS3*KO, *SKIV2L*KO*OAS3*KO*OAS1*KO A549 cells after poly(I:C) (1.0 μg/ml) treatment for 4 h. (B) Western blot analysis of RIG-I and MDA5 in single or double gene knockout cells (as indicated on top) treated with IFN-β (20 ng/ml) for 24 h. (C) RT-qPCR analysis of *IFNB1* mRNA in WT, *DDX58*KO and *IFIH1*KO after SeV infection for 24 h or EMCV infection for 6 h. Data are shown as mean ± SEM of three or four independent experiments. Two-sided Student's *t* test; **P < 0.01, ***P < 0.001; ns, not significant. *P* values = 0.007922 (Sev, WT vs *DDX58*KO), 0.182148 (Sev, WT vs *IFIH1*KO), 0.853967 (EMCV, WT vs *DDX58*KO), 0.000002 (EMCV, WT vs *IFIH1*KO). (D) RT-qPCR analysis of *IFNB1* mRNA in WT and indicated gene knockout cells after poly(I:C) treatment (0.3 μg/ml) for 4 h. Fold change of *IFNB1* mRNA compared to mock-treated WT cells is shown. Data are shown as mean ± SEM of four independent experiments. Two-sided Student's *t* test; ***P < 0.001. *P* values = 0.000007 (WT vs *DDX58*KO), <0.000001 (WT vs *SKIV2L*KO), <0.000001 (*SKIV2L*KO vs *SKIV2L*KO*DDX58*KO). (E, F) *DDX58*KO cells were treated with HMW poly(I:C) or IFN-β. Expression of ISG and IFNB1 were analyzed using RT-qPCR. Data are shown as mean ± SEM of three independent experiments. Source data are available online for this figure.

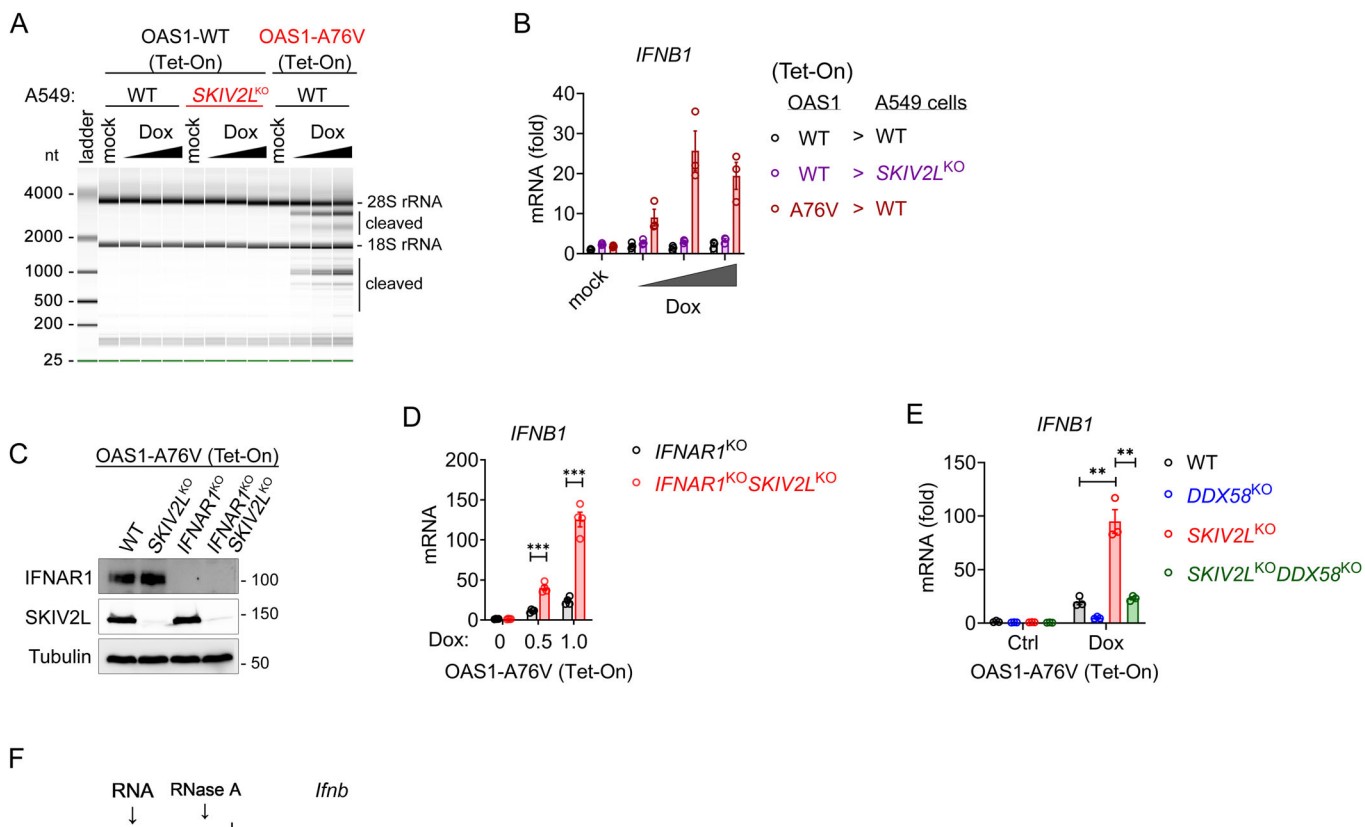

**Figure EV3. SKIV2L restricts OAS1-mediated autoinflammation.**

(A) rRNA cleavage analysis of total RNA in WT or *SKIV2L*KO cells after induction of WT or A76V mutant OAS1 with doxycycline (1.0 μg/ml) for 24 h. Data are representative of at least three independent experiments. (B) RT-qPCR analysis of *IFNB1* mRNA in WT or *SKIV2L*KO cells after induction of WT or A76V mutant OAS1 with doxycycline (1.0 μg/ml) for 24 h. Data are shown as mean ± SEM of three independent experiments. (C) Western blot analysis of IFNAR1 and SKIV2L proteins in WT and indicated knockout cells stably expressing inducible A76V mutant OAS1. (D) RT-qPCR analysis of *IFNB1* mRNA in in WT and indicated knockout cells after induction of A76V mutant OAS1 with doxycycline (0.5, 1.0 μg/ml) for 24 h. Data are shown as mean ± SEM of three independent experiments. Two-sided Student's *t* test; ***P < 0.001. *P* values = 0.000086 (Dox 0.5), 0.000036 (Dox 1.0). (E) RT-qPCR analysis of *IFNB1* mRNA in WT, *SKIV2L*KO, *DDX58*KO, *SKIV2L*KO *DDX58*KO cells after induction of OAS1 A76V mutant with doxycycline (1.0 μg/ml) for 24 h. Data are mean ± SEM of three independent experiments. Two-sided Student's *t* test; **P < 0.01. *P* values = 0.002764 (WT vs *SKIV2L*KO), 0.002985 (*DDX58*KO vs *SKIV2L*KO *DDX58*KO). (F) MEFs were transfected with RNA (1.0 μg/ml) isolated from WT and *SKIV2L*KO A549 cells after induction of OAS1 A76V mutant (dox 1.0 μg/ml for 24 h) with or without RNase A treatment. Expression of *Ifnb* was measured by RT-qPCR. Data are mean ± SEM of four independent experiments. Source data are available online for this figure.

