## [Peer Review File · The EMBO Journal]

RNA helicase SKIV2L limits antiviral defense and autoinflammation through the OAS-RNase L pathway

Nan Yan, Kun Yang, Beihua Dong, Abhishek Asthana, and Robert Silverman

Corresponding author(s): Nan Yan (nan.yan@utsouthwestern.edu), Kun Yang (Kun.Yang@UTSouthwestern.edu)

Review Timeline:

Submission Date:	7th Mar 24
Editorial Decision:	3rd Apr 24
Revision Received:	11th Jun 24
Editorial Decision:	11th Jul 24
Revision Received:	15th Jul 24
Accepted:	16th Jul 24

Editor: Ioannis Papaioannou

Transaction Report:

Dear Nan,

Thank you again for submitting your manuscript EMBOJ-2024-117207 for consideration by The EMBO Journal. It has been seen by three experts in the field, and we have received the full set of their comments, which I have already shared with you (they are included again below). I would also like to thank you for your draft point-by-point response to their comments and for your provisional revision plan, which were very helpful for us to reach a fair and balanced editorial decision on the manuscript.

The referees recognize that the study fits well within the scope of our journal and appreciate the significance of the topic and the quality of the data. However, they also identify a number of limitations in the study and the manuscript, and they raise several technical and conceptual concerns. Importantly, they suggest that some conclusions are not sufficiently supported by the available data and that further control experiments should be performed to strengthen the findings and test alternative hypotheses.

Given the referees' comments and recommendations, as well as your willingness to revise your study substantially and address their concerns, I would like to invite you to submit a revised version of the manuscript along with a detailed point-by-point response addressing all referees' comments. I should add that it is EMBO Journal policy to allow only a single round of major experimental revision, and acceptance of your manuscript will therefore depend on the completeness of your responses in this revised version. If you have any questions or comments, we can discuss further in a video call, if you like.

We generally allow three months as standard revision time (July 2, 2024). As a matter of policy, competing manuscripts published during this period will not negatively impact our assessment of the conceptual advance presented by your study. However, we request that you contact us as soon as possible upon publication of any related work, to discuss how to proceed. Should you foresee a problem in meeting this three-month deadline, please let us know in advance and we may be able to grant an extension.

Thank you for the opportunity to consider your work for publication in The EMBO Journal. I look forward to your revision.

Best regards,

Ioannis

Instructions for preparing your revised manuscript

1. When you are ready to submit the revision, please upload:

- A Word file of the manuscript text (including legends of main Figures, EV Figures and Tables). Please make sure that changes are highlighted (or "tracked") to be clearly visible.

- Individual production-quality figure files (one file per figure). When assembling your figures, please refer to our figure preparation guidelines in order to ensure proper formatting and readability in print as well as on screen:

If the data shown in a figure are obtained from n {less than or equal to} 2, please use scatter plots showing the individual data points.

- i. the name of the statistical test used to generate error bars and P values
- ii. the number (n) of independent experiments (please specify technical or biological replicates) underlying each data point (discussion of statistical methodology can be reported in the Materials and Methods section, but figure legends should contain a basic description of n , P , and the test applied)
- iii. the nature of the bars and error bars (s.d., s.e.m.).

- A point-by-point response to the referees' comments, with a detailed description of the changes made (as a word file). All

referees' concerns must be fully addressed and their suggestions taken on board. When preparing your letter of response to the referees' comments, please bear in mind that this will form part of the Review Process File and will therefore be available online to the community. Please note that you have the possibility to opt out of the transparent process at any stage prior to publication by letting the editorial office know (contact@embojournal.org); if you do opt out, the Review Process File link will point to the following statement: "No Review Process File is available with this article, as the authors have chosen not to make the review process public in this case.". For more details on our Transparent Editorial Process, please visit our website: <https://www.embopress.org/page/journal/14602075/authorguide#transparentprocess>

- Expanded View (EV) files (replacing Supplementary Information) that are collapsible/expandable online. A maximum of 5 EV Figures can be typeset. EV Figures should be cited as "Figure EV1, Figure EV2" etc. in the text, and their respective legends should be included in the manuscript file after the legends of regular figures. See detailed instructions regarding Expanded View files here:

- For the figures that you do NOT wish to display as Expanded View figures, they should be bundled together with their legends in a single PDF file called "Appendix", which should start with a short Table of Contents (including page numbers). Appendix figures should be referred to in the main text as: "Appendix Figure S1, Appendix Figure S2" etc. Please see detailed instructions here: <https://www.embopress.org/page/journal/14602075/authorguide#expandedview>

- A complete author checklist, which you can download from our author guidelines (<https://www.embopress.org/page/journal/14602075/authorguide>). Please note that the checklist will also be part of the Review Process File.

2. Please note that no statistics should be calculated and shown in Figures if $n=2$.

3. Before submitting your revision, primary datasets (and computer code, where appropriate) produced in this study need to be deposited in appropriate public databases (see <https://www.embopress.org/page/journal/14602075/authorguide#dataavailability>). The accession numbers and databases should be listed in a formal "Data availability" section (placed after Materials and Methods) that follows the model below (see also <https://www.embopress.org/page/journal/14602075/authorguide#dataavailability>):

Data availability

- RNA-seq data: Gene Expression Omnibus GSE46843 (<https://www.ncbi.nlm.nih.gov/geo/query/acc.cgi?acc=GSE46843>)
- [data type]: [name of the resource] [accession number/identifier/doi] ([URL or identifiers.org/DATABASE:ACCESSION])

*** All links should resolve to a page where the data can be accessed. ***

*** Please remember to provide in the Data availability section of your revised manuscript reviewer passwords if the datasets are not yet public. ***

*** The Data Availability Section is restricted to new primary data that are part of this study. In case you have no data that require deposition in a public database, please state so instead of referring to the database: "Our study includes no data deposited in public repositories." under the heading "Data availability". ***

4. Please check that the title and the abstract of the manuscript are brief, yet explicit, even to non-specialists. The length of the title should not exceed 100 characters, and the abstract should be a single paragraph not exceeding 175 words.

5. Please also note our reference format: <https://www.embopress.org/page/journal/14602075/authorguide#referencesformat>.

7. Please remember: digital image enhancement is acceptable practice, as long as it accurately represents the original data and conforms to community standards. If a figure has been subjected to significant electronic manipulation, this must be noted in the figure legend or in the "Materials and Methods" section. The editors reserve the right to request original versions of figures and the original images that were used to assemble the figure.

8. Our journal encourages inclusion of data citations in the reference list to directly cite datasets that were obtained from public databases. Data citations in the article text are distinct from normal bibliographical citations and should directly link to the database records from which the data can be accessed. In the main text, data citations are formatted as follows: "Data ref:

Smith et al, 2001" or "Data ref: NCBI Sequence Read Archive PRJNA342805, 2017". In the Reference list, data citations must be labeled with "[DATASET]". A data reference must provide the database name, accession number/identifiers, and a resolvable link to the landing page from which the data can be accessed at the end of the reference. Further instructions are available at: <https://www.embopress.org/page/journal/14602075/authorguide#referencesformat>.

9. We request authors to consider both actual and perceived competing interests. Please review our policy (<https://www.embopress.org/page/journal/14602075/authorguide#conflictsofinterest>) and update your competing interests statement if necessary. Please name this section 'Disclosure and competing interests statement' and place it after the Acknowledgements section.

10. Please note that all corresponding authors are required to provide an ORCID ID upon submission of a revised manuscript (<https://orcid.org/>). Please find instructions on how to link your ORCID ID to your account in our manuscript tracking system in our Author guidelines (<https://www.embopress.org/page/journal/14602075/authorguide#authorshipguidelines>).

11. We use CRediT to specify the contributions of each author in the journal submission system. CRediT replaces the author contribution section, which should be removed from the manuscript. Please use the free text box to provide more detailed descriptions. See also guide to authors: <https://www.embopress.org/page/journal/14602075/authorguide#authorshipguidelines>.

13. We would also welcome the submission of cover suggestions or motifs to be used by our Graphics Illustrator in designing a cover.

14. Please use the link below to submit your revision:
<https://emboj.msubmit.net/cgi-bin/main.plex>

Referee #1:

This study from Yang and colleagues investigates the role of RNA helicase SKIV2L in limiting the innate immune response to dsRNA stimuli through the OAS-RNase L pathway. The authors claim that SKIV2L limits the innate immune response to dsRNA stimuli by restricting the availability of RNase L-cleaved RNA through RNA exosome-mediated degradation. Subsequently, in the absence of SKIV2L, RNase L-processed RNA accumulates in the system activating the RIG-I/MDA5-MAVS signaling pathway and enhancing the production of type I interferon to potentially induce autoimmunity. However, despite providing results from various human and mouse knockout cells, this manuscript suffers from major technical and conceptual problems.

Major conceptual points:

1. The authors use OAS3 as the primary producer of 2-5A based of previous A549 cell-based studies. However, recent findings (PMID: 35835913, 34581622, 34342578) suggest OAS1 to be another major producer of 2-5A that through RNase L pathway inhibit SARS-CoV-2 replication. Therefore, in addition to OAS3, the contribution of OAS1 needs to be addressed.
2. One curious issue the authors did not seem to examine the possibility that SKIV2L and the RNA exosome may be directly degrading external dsRNA or pIC PAMP thereby reducing RLR-MAVS sensing. A similar mechanism is well-established for dsDNA sensing and mediated through TREX1.
3. In Fig. 3, the authors use SKIV2L and RLR double knockout cells to make a point that SKIV2L contributes to reducing OAS-RNase L mediated innate immune activation, assuming that OAS-RNase L mediated RNA processing is upstream of RLR sensing. However, this data is equally compatible with a model that does not require any "processed RNA sensing". Conceptually, using double knockouts do not generally provide evidence for direct signaling connection.
4. According to the authors model the dsRNA PAMP pIC is being sensed by OAS, cleaved by RNase L and these products are being sensed through RLR. However, no direct proof of the ligand is provided to support this model.
5. In previous publication it was indicated that cellular RNAs processed by RNase L acts as ligand for RLR pathway (Malathi et al 2007). In that case one would expect RNA cleavage pattern to be different between WT and SKIV2L-KO cells, which does not show any change in Fig. 4D or E?

Major technical points:

1. The authors used various CRISPR-KO cell lines. However, given the propensity of generating multiple mutations one needs to establish the authenticity of the approach using rescue of the targeted protein and reversal of the phenotype.
2. Synthetic dsRNA, besides pIC needs to be included in critical assays.
3. Why there is 10 fold differences in IFN β induction between Fig. 3D/E and 3E?

4. Please include KO cell characterization in supplementary results.
5. It's not clear how apoptosis is induced in SKIV2L-deficient cells.

Minor issues:

1. Line 63, It should be IFN 'production', not 'response'.
2. In the text, Fig 3H comes before 3G. It would be better if these were in chronological order.
3. pIC concentrations used needs to be mentioned in the figure legends.
4. The sentence " OAS recognizes viral RNA and then activates RNase L. RNase L then cleaves both cellular and viral RNA, which subsequently activates RIG-I/MDA5." Sounds like it is an established fact, which it is not given that OAS and RNase L KO cells are capable of inducing IFN by dsRNA. Please correct.
5. Line 28: In human there are 4 members of OAS family (OAS1-3 and OASL).
6. Line 112: "These data importantly demonstrate that the OAS-RNase L pathway has tremendous 'power' for amplifying RNA-induced IFN response, much higher than direct sensing of exogenous RNA (as in WT cells), but most of that 'power' is restricted by SKIV2L. Please correct with appropriate language.

Referee #2:

Innate immune signaling is carefully tuned to allow sensitive detection of infections while avoiding autoinflammation. This manuscript adds to our understanding of how this 'tuning' is achieved molecularly. The authors' discovery that SKIV2L blocks interferon and cell death responses upon OAS-RNaseL activation is important, novel and of broad interest. The data are of high quality and the manuscript is well written and beautifully laid out. This reviewer supports publication in EMBO J if the following specific points can be addressed.

Major points

1. The claim of 'processed RNA sensing' (text and Figs 3H and 6G) is not fully supported by the data and may be misleading. Enhanced IFN β induction in SKIV2L KO cells triggered by Poly I:C (Figs 1&3) and SINV (Fig 5) could be due to a priming effect in SKIV2L KO cells. For example, it is possible that RIG-I (encoded by an ISG) is expressed at higher levels in SKIV2L KO cells due to increased tonic interferon signalling. In this scenario, 'processed RNA sensing' would be only relevant to setting elevated RIG-I baseline levels, with higher RIG-I expression explaining the enhanced response to Poly I:C or SINV. The authors need to test this possibility by:

- 1a. analyzing the expression of RIG-I, MDA5, MAVS and ideally also of IRF3 and TBK1 in SKIV2L KO cells using western blot and RT qPCR
- 1b. analyzing the baseline transcriptome in SKIV2L KO cells using RNAseq, with a focus on ISGs in data analysis
- 1c. generating SKIV2L IFNAR double KO cells and comparing Poly I:C and/or SINV triggered IFN induction to IFNAR single KO cells

2. The MAVS-dependent IFN induction Fig 6B-D is an important observation because no foreign RNA is introduced, potentially supporting the 'processed RNA sensing' hypothesis. To substantiate this, the authors should:

- a. confirm that induction of OAS1 A76V results in 2-5A production and/or rRNA cleavage, using assays shown in Fig 4
- b. include OAS1 WT as a control
- c. determine whether the phenotype in Fig 6C/D is RNase L- and RIG-I-dependent but IFNAR-independent (using double KOs)

Minor points

3. Fig 1F. Is mouse TTC37 destabilized in iSkiv2l^{-/-} cells?
4. Fig 2/3. Do the authors have western blot data for full-length Casp-3 and PARP and for total PKR and eIF2 α ?
5. Fig 3F. Functional validation of RIG-I and MDA5 knockout cells using SeV and EMCV infections (that activate RIG-I and MDA5, respectively) would be desirable.
6. Fig 4A. With n=2, individual data points need to be shown.
7. Fig 4B/C. It would help the reader to arrange these panels such that they show the same order of KOs from left to right.
8. Line 146. Panel C is missing in Fig 5.
9. Line 158. Please call out in text Fig 6D.
10. The discussion of Eckard et al 2014 could be expanded. E.g. Eckard et al suggested divergent roles of TTC37 and SKIV2L, which appears at odds with this study.

Referee #3:

This manuscript by Yang et al. describes how the SKIV2L exosome complex restricts the capacity of the OAS-RNase L pathway to activate the RIG-I-MAVS pathway in response to endogenous or exogenous dsRNA. The authors find that loss of SKIV2L leads to increased type I IFN induction in response to poly(I:C) in A549 and in primary skin fibroblasts. In addition, they find that SKIV2L deficiency enhances apoptosis in response to poly(I:C) stimulation in a manner that is dependent on SKIV2L helicase activity. They find that the above observations are mediated via the OAS-RNase L and the RIG-I-MAVS pathway (but not the PKR pathway). In addition, they report that SKIV2L KO A549 have increased resistance to Sindbis infection due to an increased IFN response. Finally, the authors elegantly demonstrate that loss of SKIV2L increases a spontaneous IFN response in the context of an OAS gain-of-function mutation. This manuscript is written in a transparent manner. The experiments are well executed and convincing. A small concern is the novelty of the study: an earlier paper (Eckard et al) already demonstrated that SKIV2L deficiency increases the response to exogenous dsRNA. However, this was attributed to a different underlying mechanism, hence the study by Yang et al. is relevant to the field and gives these earlier findings a different angle. Overall, the study fits well within the scope of EMBO Journal and I only have a few outstanding issues that should be addressed to make this paper suitable for publication in the EMBO Journal. Most importantly, the authors could make use of the SKIV2L-deficient mice or the patient cells with SKIV2L mutations, both of which they have at their disposal to strengthen their findings.

Major points:

- Fig. 1B: these findings would be strengthened if the enhanced IFN response in SKIV2L KO cells would be demonstrated at protein level as well (e.g. by immunoblot analysis using antibodies directed against an ISG).
- Fig. 1: Is the poly(I:C) that is used high-molecular weight (HMW) or low-molecular weight (LMW) poly(I:C)? I assume it is LMW since the poly(I:C) response is completely RIG-I dependent (Fig. 3B). Does stimulation with HMW poly(I:C) or another MDA5 ligand similarly trigger an increased IFN response in SKIV2L KO cells?
- Fig. 1B: the authors do not observe a spontaneous IFN response in the SKIV2L-deficient cells (i.e. in absence of an RLR agonist). Could this be explained by low expression levels of OAS or RIG-I in the absence of IFNAR signaling (i.e. in the absence of the positive feedback loop)? The authors could test whether stimulation with recombinant type I IFN triggers spontaneous IFN-beta transcription in SKIV2L KO cells (a trick that is used by the authors in Fig. 6E for ADAR1 deficient cells).
- Fig. 2D: does SKIV2L WT (but not E424Q) overexpression also rescue the increased IFN response upon poly(I:C) treatment? This is important because the entire paper is based on 2 SKIV2L KO clones only.
- Fig. 3E: How do the authors explain the MAVS-independent IFN response in SKIV2L KO cells? Both 'direct' and 'processed' RNA sensing would presumably involve MAVS.
- Fig. 3F/G: Can the authors confirm that the RIG-I KO cells still have an intact IFN pathway (i.e. do the cells respond to treatment with an MDA5 agonist or recombinant type I IFN)?
- The authors convincingly demonstrate that SKIV2L KO cells have increased resistance to infection with Sindbis virus due to increased IFN-beta transcription. Since the authors previously generated SKIV2L-deficient mice (Fig. 1E and Yang et al. J. Clinical Investigation 2022), they could easily investigate whether these mice display increased resistance to Sindbis infection. Irrespective of the outcome of such an experiment, this would reveal the relative contribution of the SKIV2L complex to antiviral defense in vivo, especially in light of the argument that the authors are making in the discussion with regards to the potential of SKIV2L inhibitors as novel antiviral compounds.
- The authors find that the expression of an OAS GoF variant induces an increased IFN response in SKIV2L KO cells. SKIV2L loss-of-function mutations were previously reported and are associated with THES. Does the expression of the SKIV2L LoF variant phenocopy the observations in the SKIV2L KO cells? The authors previously obtained cells from a patient with THES2 due to SKIV2L mutations. Can the authors recapitulate some of their findings (e.g. increased IFN upon poly(I:C), increased viral resistance) in these patient cells?

Minor points:

- Figure referrals in line 127 and 128 are incorrect (3D and 3E should be 4D and 4E).

We would like to thank the editor and reviewers for spending valuable time evaluating our manuscript. We have performed additional experiments and revised the manuscript to address concerns and suggestions raised during the review. Text changes in the manuscript are highlighted in blue. Below, we provide a point-by-point response to each comment.

Referee #1 (Report for Author)

This study from Yang and colleagues investigates the role of RNA helicase SKIV2L in limiting the innate immune response to dsRNA stimuli through the OAS-RNase L pathway. The authors claim that SKIV2L limits the innate immune response to dsRNA stimuli by restricting the availability of RNase L-cleaved RNA through RNA exosome-mediated degradation. Subsequently, in the absence of SKIV2L, RNase L-processed RNA accumulates in the system activating the RIG-I/MDA5-MAVS signaling pathway and enhancing the production of type I interferon to potentially induce autoimmunity. However, despite providing results from various human and mouse knockout cells, this manuscript suffers from major technical and conceptual problems.

Major conceptual points:

1. The authors use OAS3 as the primary producer of 2-5A based on previous A549 cell-based studies. However, recent findings (PMID: 35835913, 34581622, 34342578) suggest OAS1 to be another major producer of 2-5A that through RNase L pathway inhibits SARS-CoV-2 replication. Therefore, in addition to OAS3, the contribution of OAS1 needs to be addressed.

Response: We agree OAS1 can also produce 2-5A during viral infection, and its overexpression dramatically suppresses SARS-CoV-2 replication (PMID: 34581622, 34342578). We used OAS3-KO A549 cells obtained from Susan Weiss, who previously demonstrated that the predominant OAS activity in A549 cells is from OAS3 (PMID: 26858407, panel B in the figure below). We did observe residual poly(I:C)-induced cell death in *SKIV2L*^{KO}*OAS3*^{KO} cells in **Figure 3B**, suggesting a role for other OASs. To determine the contribution of OAS1, we further knocked out OAS1 and generated *SKIV2L*^{KO}*OAS3*^{KO}*OAS1*^{KO} triple knockout cells. Compared with *SKIV2L*^{KO}*OAS3*^{KO}, *SKIV2L*^{KO}*OAS3*^{KO}*OAS1*^{KO} showed a further reduction in poly(I:C)-induced cell death (**Figure EV2A** in revised manuscript), suggesting a role for OAS1 in this assay.

2. One curious issue the authors did not seem to examine the possibility that SKIV2L and the RNA exosome may be directly degrading external dsRNA or pIC PAMP thereby reducing RLR-MAVS sensing. A similar mechanism is well-established for dsDNA sensing and mediated through TREX1.

Response: We appreciate this insightful comment. After poly(I:C) stimulation, we observed no major difference in the activation of the PKR-eIF2 α pathway (measured by phosphorylation of PKR and eIF2 α) between WT and SKIV2L KO cells (Fig. 2A and Fig. 3A lane 5 versus lane 6). These data suggest that SKIV2L RNA exosome does not substantially degrade incoming external dsRNA, as such degradation would likely impair the PKR-eIF2 α RNA sensing pathway.

3. In Fig. 3, the authors use SKIV2L and RLR double knockout cells to make a point that SKIV2L contributes to reducing OAS-RNase L mediated innate immune activation, assuming that OAS-RNase L mediated RNA processing is upstream of RLR sensing. However, this data is equally compatible with a model that does not require any "processed RNA sensing". Conceptually, using double knockouts do not generally provide evidence for direct signaling connection.

Response: We apologize for the confusion. We agree that Fig. 3G (now Fig. EV2D in the revised manuscript) alone, using SKIV2L and RLR double knockout cells, does not provide sufficient evidence for direct signaling. Our conclusion regarding "processed RNA sensing" and "direct RNA sensing" is primarily drawn from Fig. 3D, where we used SKIV2L and *RNase L* double knockout cells that blocks RNA processing by RNase L. This approach specifically demonstrates the role of RNase L in "RNA processing" upstream of RLR sensing. The original Fig. 3G, which utilized SKIV2L and RLR double knockout cells, was intended to show which RNA sensor (RIG-I or MDA5) mediates the enhanced IFN response in SKIV2L KO cells. This data helps to identify the specific RNA sensor involved rather than establishing a direct signaling connection. We have clarified this point in the revised manuscript.

4. According to the authors model the dsRNA PAMP pIC is being sensed by OAS, cleaved by RNase L and these products are being sensed through RLR. However, no direct proof of the ligand is provided to support this model.

Response: We appreciate the reviewer's comment. Our data do not support the idea that dsRNA PAMP poly(I:C) *per se* is cleaved by RNase L. Instead, our data suggests that activated RNase L cleaves cellular RNAs, generating endogenous ligands for the RLR pathway. In the revised manuscript, we used human A549 cells expressing OAS-GoF mutant to naturally activate RNase L without introducing external dsRNA, then isolated cellular RNA from these cells and transfected it into a wild-type MEF cells. The RNA from SKIV2L-KO exhibited increased immunostimulatory activity, consistent with our model that SKIV2L RNA exosome degrades RNaseL-cleaved RNA (**Fig. EV3F**). This experiment provides evidence that the immunostimulatory RNA ligands are generated from RNase L cleavage of cellular RNA.

5. In previous publication it was indicated that cellular RNAs processed by RNase L acts as ligand for RLR pathway (Malathi et al 2007). In that case one would expect RNA cleavage pattern to be different between WT and SKIV2L-KO cells, which does not show any change in Fig. 4D or E?

Response: Thank you for this important observation. The rRNA cleavage assay using the Bioanalyzer in Fig. 4D or 4E was used as a readout for RNase L enzymatic activity. This assay does not capture the small RNA species that are potential RIG-I ligands (as defined by Bob Silverman, PMID: 17653195). The lack of a major difference in rRNA cleavage between WT and SKIV2L-KO cells indicates that SKIV2L KO does not affect the overall RNase L enzymatic activity.

As mentioned in our response to the previous comment, we found that RNA isolated from SKIV2L-KO cells expressing the OAS1-A76V (gain-of-function) indeed induced more IFN production compared to RNA from WT cells (**Fig. EV3F**). This suggests more immunostimulatory RNA accumulates in SKIV2L-KO cells.

Major technical points:

1. The authors used various CRISPR-KO cell lines. However, given the propensity of generating multiple mutations one needs to establish the authenticity of the approach using rescue of the targeted protein and reversal of the phenotype.

Response: We agree that demonstrating the authenticity of the CRISPR-KO approach through rescue experiments is essential. In the revised manuscript, we included the rescue experiments of target genes. We showed that the key phenotypes of our study were reversed after re-expression of targeted protein (**Fig. 1C and 1D, Fig. 2D**).

2. Synthetic dsRNA, besides pIC needs to be included in critical assays.

Response: We have included 5' triphosphate hairpin RNA (3p-hpRNA), generated by in vitro transcription of an influenza A (H1N1) virus sequence, in our critical assays. We observed similar phenotypes to those seen with poly(I:C) (**Fig. EV1B**). This confirms that the observed effects are not specific to poly(I:C) but are generalizable to other synthetic dsRNAs.

3. Why there is 10 fold differences in IFN β induction between Fig. 3D/E and 3E?

Response: The differences in IFN- β induction between the original Fig. 3D/E and 3G (now Fig. EV2D in the revised manuscript) are due to variations in experimental conditions. Specifically, in Fig. 3D/E, we used poly(I:C) at a concentration of 1.0 $\mu\text{g/ml}$, whereas in Fig. 3G, we used a concentration of 0.3 $\mu\text{g/ml}$. These differences in poly(I:C) concentration account for the observed variation in IFN- β induction levels. The concentrations used in each stimulation were added in the revised figure legends.

4. Please include KO cell characterization in supplementary results.

Response: All KO cell lines used in this study were confirmed by Western blot. Furthermore, key phenotypes were validated through rescue experiments of the targeted protein.

5. It's not clear how apoptosis is induced in SKIV2L-deficient cells.

Response: Our data suggest that RNase L mediates apoptosis in SKIV2L-deficient cells following dsRNA stimulation (Fig. 3A). RNase L is a well-established innate immune pathway that induces apoptosis (PMID 14570908, 15604285, 9325243, 31330998, 26263979, 11152576, 14583476, 10200477). However, the molecular mechanism by which RNase L induces apoptosis remains elusive. A recent study from Bob Silverman indicates ZAK α -dependent ribotoxic stress response (PMID: 38551960).

Minor issues:

1. Line 63, It should be IFN 'production', not 'response'.
2. In the text, Fig 3H comes before 3G. It would be better if these were in chronological order.
3. pIC concentrations used needs to be mentioned in the figure legends.
4. The sentence "OAS recognizes viral RNA and then activates RNase L. RNase L then cleaves both cellular and viral RNA, which subsequently activates RIG-I/MDA5." Sounds like it is an established fact, which it is not given that OAS and RNase L KO cells are capable of inducing IFN by dsRNA. Please correct.
5. Line 28: In human there are 4 members of OAS family (OAS1-3 and OASL).
6. Line 112: "These data importantly demonstrate that the OAS-RNase L pathway has tremendous 'power' for amplifying RNA-induced IFN response, much higher than direct sensing of exogenous RNA (as in WT cells), but most of that 'power' is restricted by SKIV2L. Please correct with appropriate language.

Response: We fixed all minor issues above in the revised manuscript.

Referee #2 (Report for Author)

Innate immune signaling is carefully tuned to allow sensitive detection of infections while avoiding autoinflammation. This manuscript adds to our understanding of how this 'tuning' is achieved molecularly. The authors' discovery that SKIV2L blocks interferon and cell death responses upon OAS-RNaseL activation is important, novel and of broad interest. The data are of high quality and the manuscript is well written and beautifully laid out. This reviewer supports publication in EMBO J if the following specific points can be addressed.

Major points

1. The claim of 'processed RNA sensing' (text and Figs 3H and 6G) is not fully supported by the data and may be misleading. Enhanced IFN β induction in SKIV2L KO cells triggered by Poly I:C (Figs 1&3) and SINV (Fig 5) could be due to a priming effect in SKIV2L KO cells. For example, it is possible that RIG-I (encoded by an ISG) is expressed at higher levels in SKIV2L KO cells due to increased tonic interferon signalling. In this scenario, 'processed RNA sensing' would be only relevant to setting elevated RIG-I baseline levels, with higher RIG-I expression explaining the enhanced response to Poly I:C or SINV. The authors need to test this possibility by:

1a. analyzing the expression of RIG-I, MDA5, MAVS and ideally also of IRF3 and TBK1 in SKIV2L KO cells using western blot and RT qPCR

Response: To investigate whether SKIV2L KO cells exhibit increased tonic IFN and ISGs (including RIG-I-like receptors), thus priming cells for enhanced RNA sensing, we evaluated the expression of molecules involved

in the RNA sensing pathway at both the protein and mRNA levels. Our analysis revealed no major differences between WT and SKIV2L KO cells in the expression of these molecules (**Fig. EV1D and 1E**).

1b. analyzing the baseline transcriptome in SKIV2L KO cells using RNAseq, with a focus on ISGs in data analysis

Response: To assess baseline IFN signaling and ISG expression in SKIV2L KO cells, we conducted qPCR analysis of a broad panel of ISGs. Our results revealed no overall increase in tonic ISG expression in SKIV2L KO cells compared to wild-type cells (**Fig. EV1F**).

1c. generating SKIV2L IFNAR double KO cells and comparing Poly I:C and/or SINV triggered IFN induction to IFNAR single KO cells.

Response: We generated *IFNAR1*^{KO} single knockout and *IFNAR1*^{KO}*SKIV2L*^{KO} double knockout cells (**Fig. EV1G**). Compared to *IFNAR1*^{KO} cells, *IFNAR1*^{KO}*SKIV2L*^{KO} cells still exhibited enhanced IFN response after dsRNA stimulation or SINV infection (**Fig. EV1H**).

Collectively, these findings provide different lines of evidence ruling out the possibility that *SKIV2L*^{KO} elevates baseline ISG, such as RIG-I, therefore enhancing RNA sensing signaling.

2. The MAVS-dependent IFN induction Fig 6B-D is an important observation because no foreign RNA is introduced, potentially supporting the 'processed RNA sensing' hypothesis. To substantiate this, the authors should:

a. confirm that induction of OAS1 A76V results in 2-5A production and/or rRNA cleavage, using assays shown in Fig 4.

Response: We measured cellular 2-5A production after induction of OAS1 A76V gain-of-function mutant using FRET assay (**Fig. 6B**) and rRNA cleavage using Bioanalyzer (**Fig. 6C**).

b. include OAS1 WT as a control

Response: We included inducible OAS1 WT as control in our experiment. We found that induction of WT OAS1 did not result in rRNA cleavage (**Fig. EV3A**) or IFN induction (**Fig. EV3B**).

c. determine whether the phenotype in Fig 6C/D is RNase L- and RIG-I-dependent but IFNAR-independent (using double KOs)

Response: We generated SKIV2L/RNase L, SKIV2L/RIG-I, SKIV2L/IFNAR double KO carrying an inducible OAS1 A76V GoF mutant. Our findings demonstrate that further deletion of RNase L or RIG-I blocked IFN induction in OAS1 A76V-expressing *SKIV2L*^{KO} cells (**Fig. 6F, Fig. EV3E**). Furthermore, compared to IFNAR single KO, SKIV2L/IFNAR double KO still showed increased IFN expression after induction of OAS1 A76V

mutant (**Fig. EV3C and EV3D**). These results suggest that it is unlikely that *SKIV2L*^{KO} enhanced IFN response through elevated tonic IFN signaling.

Minor points

3. Fig 1F. Is mouse TTC37 destabilized in iSkiv2l^{-/-} cells?

Response: We further measured TTC37 in iSkiv2l^{-/-} mouse skin-derived fibroblasts using Western blot and found a reduced protein level of TTC37 (**Fig. 1H**). This observation, combined with the results obtained from human SKIV2L KO A549 cells, leads us to conclude that the SKI complex is destabilized when SKIV2L is lost in either human or mouse cells.

4. Fig 2/3. Do the authors have western blot data for full-length Casp-3 and PARP and for total PKR and eIF2a?

Response: We have included western blot data for total PKR and total eIF2a in the revised figures. The antibodies we used for Casp-3 and PARP are specific for cleaved ones and do not detect full-length.

5. Fig 3F. Functional validation of RIG-I and MDA5 knockout cells using SeV and EMCV infections (that activate RIG-I and MDA5, respectively) would be desirable.

Response: We infected *DDX58*^{KO} and *IFIH1*^{KO} A549 cells with SeV or EMCV and measured IFN mRNA expression post infection to functionally validate both KO cells. Specifically, IFN induction was blocked in *DDX58*^{KO} A549 cells after SeV infection and in *IFIH1*^{KO} cells after EMCV infection. It's worth noting that we observed weak IFN induction in SeV-infected A549 cells, probably due to their low expression of TMPRSS2 protease (PMID: 32165541), which is required for efficient SeV infection (PMID: 23966399).

6. Fig 4A. With n=2, individual data points need to be shown.

Response: Individual data points were shown in the revised Figure 4A.

7. Fig 4B/C. It would help the reader to arrange these panels such that they show the same order of KOs from left to right.

Response: Two panels were re-arranged in the same order in the revised Figure 4.

8. Line 146. Panel C is missing in Fig 5.

Response: Fixed.

9. Line 158. Please call out in text Fig 6D.

Response: Fixed.

10. The discussion of Eckard et al 2014 could be expanded. E.g. Eckard et al suggested divergent roles of

TTC37 and SKIV2L, which appears at odds with this study.

Response: We expanded discussion of the Eckard et al 2014 study.

Referee #3 (Report for Author)

This manuscript by Yang et al. describes how the SKIV2L exosome complex restricts the capacity of the OAS-RNase L pathway to activate the RIG-I-MAVS pathway in response to endogenous or exogenous dsRNA. The authors find that loss of SKIV2L leads to increased type I IFN induction in response to poly(I:C) in A549 and in primary skin fibroblasts. In addition, they find that SKIV2L deficiency enhances apoptosis in response to poly(I:C) stimulation in a manner that is dependent on SKIV2L helicase activity. They find that the above observations are mediated via the OAS-RNase L and the RIG-I-MAVS pathway (but not the PKR pathway). In addition, they report that SKIV2L KO A549 have increased resistance to Sindbis infection due to an increased IFN response. Finally, the authors elegantly demonstrate that loss of SKIV2L increases a spontaneous IFN response in the context of an OAS gain-of-function mutation. This manuscript is written in a transparent manner. The experiments are well executed and convincing. A small concern is the novelty of the study: an earlier paper (Eckard et al) already demonstrated that SKIV2L deficiency increases the response to exogenous dsRNA. However, this was attributed to a different underlying mechanism, hence the study by Yang et al. is relevant to the field and gives these earlier findings a different angle. Overall, the study fits well within the scope of EMBO Journal and I only have a few outstanding issues that should be addressed to make this paper suitable for publication in the EMBO Journal. Most importantly, the authors could make use of the SKIV2L-deficient mice or the patient cells with SKIV2L mutations, both of which they have at their disposal to strengthen their findings.

Major points:

- Fig. 1B: these findings would be strengthened if the enhanced IFN response in SKIV2L KO cells would be demonstrated at protein level as well (e.g. by immunoblot analysis using antibodies directed against an ISG).

Response: We have included an immunoblot analysis of an ISG, RSAD2, as a readout of IFN response at protein level (Fig. EV1C).

- Fig. 1: Is the poly(I:C) that is used high-molecular weight (HMW) or low-molecular weight (LMW) poly(I:C)? I assume it is LMW since the poly(I:C) response is completely RIG-I dependent (Fig. 3B). Does stimulation with HMW poly(I:C) or another MDA5 ligand similarly trigger an increased IFN response in SKIV2L KO cells?

Response: The poly(I:C) used in our initial manuscript is low-molecular weight (LMW). In the revised manuscript, we included HMW stimulation and also observed an increased IFN response in SKIV2L KO cells (Fig. EV1A).

- Fig. 1B: the authors do not observe a spontaneous IFN response in the SKIV2L-deficient cells (i.e. in absence of an RLR agonist). Could this be explained by low expression levels of OAS or RIG-I in the absence

of IFNAR signaling (i.e. in the absence of the positive feedback loop)? The authors could test whether stimulation with recombinant type I IFN triggers spontaneous IFN-beta transcription in SKIV2L KO cells (a trick that is used by the authors in Fig. 6E for ADAR1 deficient cells).

Response: We treated SKIV2L KO cells with recombinant IFN- β and did not observed spontaneous IFN- β in SKIV2L KO cells, although the expression of DDX58, IFIH1 and OAS3 were dramatically upregulated (**Fig. EV1I**).

- Fig. 2D: does SKIV2L WT (but not E424Q) overexpression also rescue the increased IFN response upon poly(I:C) treatment? This is important because the entire paper is based on 2 SKIV2L KO clones only.

Response: We transduced two lines of SKIV2L KO cells with either WT SKIV2L or catalytically inactive E424Q mutant. Our findings indicate that only WT SKIV2L rescued the increased IFN response upon dsRNA stimulation (**Fig. 1C and 1D**), suggesting that the enhanced IFN response was indeed caused by loss of SKIV2L.

- Fig. 3E: How do the authors explain the MAVS-independent IFN response in SKIV2L KO cells? Both 'direct' and 'processed' RNA sensing would presumably involve MAVS.

Response: In addition to RIG-I like receptors, A549 cells also express the dsRNA receptor TLR3, which mediates IFN response after viral infection or poly(I:C) treatment (PMID: 25880109, 19234180, 15778392, 15731229). Therefore, we speculate that IFN response in SKIV2L/MAVS DKO cells is attributed to RLR-independent RNA sensing pathway, such as TLR3.

- Fig. 3F/G: Can the authors confirm that the RIG-I KO cells still have an intact IFN pathway (i.e. do the cells respond to treatment with an MDA5 agonist or recombinant type I IFN)?

Response: We treated RIG-I KO cells with recombinant IFN- β or HMW poly(I:C) and still observed induction of ISG and type I IFN, suggesting RIG-I KO cells still have an intact IFN pathway (**Fig. EV2E and EV2F**).

- The authors convincingly demonstrate that SKIV2L KO cells have increased resistance to infection with Sindbis virus due to increased IFN-beta transcription. Since the authors previously generated SKIV2L-deficient mice (Fig. 1E and Yang et al. J. Clinical Investigation 2022), they could easily investigate whether these mice display increased resistance to Sindbis infection. Irrespective of the outcome of such an experiment, this would reveal the relative contribution of the SKIV2L complex to antiviral defense in vivo, especially in light of the argument that the authors are making in the discussion with regards to the potential of SKIV2L inhibitors as novel antiviral compounds.

Response: We tested Sindbis virus intraperitoneal infection in postnatal whole-body inducible *Skiv2* KO mice (germ-line whole-body KO is early embryonic lethal) and found moderate resistance to Sindbis virus infection (see figure below). However, it's important to note that these *iSkiv2*^{-/-} mice already have severe skin

inflammation and perturbation of immune cells including T cells and B cells, as described in our previous publication (Yang et al. JCI 2022), which are apparent confounding factors when interpreting the data. We didn't include the data in our manuscript but would like to share with the reviewers.

- The authors find that the expression of an OAS GoF variant induces an increased IFN response in SKIV2L KO cells. SKIV2L loss-of-function mutations were previously reported and are associated with THES. Does the expression of the SKIV2L LoF variant phenocopy the observations in the SKIV2L KO cells? The authors previously obtained cells from a patient with THES2 due to SKIV2L mutations. Can the authors recapitulate some of their findings (e.g. increased IFN upon poly(I:C), increased viral resistance) in these patient cells?

Response: We restored SKIV2L expression in SKIV2L-deficiency patient-derived fibroblasts with stable retroviral transduction and used them as an "isogenic" control (**Fig. 1J**). Compared with SKIV2L-rescued cells, SKIV2L-deficient patient cells showed increase IFN response after dsRNA stimulation (**Fig. 1K**). Additionally, we observed reduced viral titer of SKIV2L-deficient patient's fibroblasts after SINV infection compared to SKIV2L-rescued cells (**Fig. 5C**), suggesting their increased viral resistance.

Minor points:

- Figure referrals in line 127 and 128 are incorrect (3D and 3E should be 4D and 4E).

Response: Fixed.

Dear Nan,

Thank you for the submission of your revised manuscript to The EMBO Journal and your patience during peer review. We have now received the comments of all three referees who assessed the revised version of your manuscript (their reports are included below). As you will see, they are all satisfied with the revision, find the manuscript substantially improved with the addition of new data, explain that the majority of their previously raised concerns have been sufficiently addressed, and now all support publication in The EMBO Journal pending a minor revision during which we would kindly ask you to address all of their remaining minor concerns regarding the need for further discussion, clarification, and improvement of data presentation. Please also include in your resubmission a point-by-point response to their comments explaining in detail all new changes to the manuscript.

From the editorial side, there are also a few changes and corrections that we need from you before we can proceed with acceptance of the manuscript for publication:

- The Figures should be removed from the main manuscript file and only be uploaded to our manuscript tracking system separately as individual Figure files. The legends of main and EV Figures should remain in the manuscript, after the list of References, with the headings "Figure legends" and "EV Figure legends", as appropriate.
- Please include a list of up to 5 keywords after the Abstract of your revised manuscript.
- The Materials and Methods need to be described in the manuscript using our "Structured Methods" format, which is now required for all research articles. According to this format, the Materials and Methods section includes a single "Reagents and Tools Table" -listing key reagents, experimental models, software and relevant equipment and including their sources and relevant identifiers- followed by a "Methods and Protocols" section describing the methods using a step-by-step protocol format. The aim is to facilitate adoption of the methodologies across labs. More information on this format as well as detailed instructions, examples, and a template (.docx) for the "Reagents and Tools Table" can be found in our author guide: <https://www.embopress.org/page/journal/14602075/authorguide#structuredmethods>.
- Please note that a "Data availability" statement (after the Materials and Methods section) is mandatory. If your study does not include newly generated datasets that need to be deposited in an external repository, please add the statement "This study includes no data deposited in external repositories." after the heading "Data availability".
- Please note that a "Disclosure and competing interests statement" (after the "Data availability" section) is mandatory. Please find more information in our author guide: <https://www.embopress.org/page/journal/14602075/authorguide#conflictsofinterest>.
- The author contributions statement should be removed from the manuscript file. Instead, we now use CRediT to specify the contributions of each author in the journal submission system. Please feel free to use the free text box to provide more detailed descriptions during submission. See also our guide to authors for more information: <https://www.embopress.org/page/journal/14602075/authorguide#authorshippinguidelines>.
- There is a callout for Figure S1A (on page 9 of the manuscript), but such a Figure does not exist. Please check and correct as appropriate.
- Please include in your resubmission a completed Author Checklist, which you can download from our author guidelines (<https://www.embopress.org/page/journal/14602075/authorguide>). Please note that the checklist will also be part of the Peer Review File (see below for more information).
- Please combine the Source Data for all EV Figures in a single master (zip) folder named "EV Figure Source Data".
- Please note that EMBO press papers are accompanied online by:
 - A) a short (2 sentences) summary of the findings and their significance,
 - B) 2-5 short bullet points highlighting the key results, and
 - C) a synopsis image in .jpg or .png format that is exactly 550 pixels wide and 300-600 pixels high (the height is variable). Please note that the text needs to be legible at the final size. Please upload this information along with your revised manuscript (the text for A and B should be provided in a separate Word file).
- Please define the annotated p values ***/** as well as provide the exact p-values for the same in the legends of Figures EV 1h; EV 2c; EV 3d as appropriate.
- Please note that the exact p values are not provided in the legends of Figures 1b, d, f, i, k; 2c; 3d-e; 5a-c; 6f-g; EV 1a-b; EV 2d; EV 3e.

- Please indicate the statistical test used for data analysis in the legends of Figures EV 1h; EV 2c; EV 3d.
- Please note that in Figure 4c there is a mismatch between the annotated p values in the Figure legend and the annotated p values in the Figure file that should be corrected.
- Please note that information related to "n" is missing in the legends of Figures 1b, d, f, i, k.

We look forward to seeing a final version of your manuscript as soon as possible. Please use this link to submit your revision: <https://emboj.msubmit.net/cgi-bin/main.plex>

Best regards,

Ioannis

Referee #1:

In this revised manuscript the authors have addressed some of my concerns and added substantial new results. Therefore, I'm supportive of its publication in EMBO J. However, some of the explanations provided by the authors needs further discussion. For example, in response to my previous comment #2 the authors states: "After poly(I:C) stimulation, we observed no major difference in the activation of the PKR-eIF2 pathway (measured by phosphorylation of PKR and eIF2) between WT and SKIV2L KO cells (Fig. 2A and Fig. 3A lane 5 versus lane 6). These data suggest that SKIV2L RNA exosome does not substantially degrade incoming external dsRNA, as such degradation would likely impair the PKR-eIF2 α RNA sensing pathway." But, the nature of RNA ligands that activate PKR (dsRNA of certain length) vs RLR pathways (ds and structured ssRNA with 5' phosphates) are quite different. Thus, similar PKR activity doesn't really answer this question. In #4 the authors used RNA isolated from human A549 cells expressing OAS-GoF mutant to naturally activate RNase L without introducing external dsRNA, then isolated cellular RNA from these cells and transfected it into a wild-type MEF cells showing the RNA from SKIV2L-KO exhibited increased immunostimulatory activity, consistent with the proposed model. However, the reason for why these processed RNA don't activate RLR signaling in the same cell, and requires transfection is not elaborated. Another point is the nature of RLR activation and the ligand necessary have been studied extensively, well beyond PMID: 17653195. Please see studies from Sun Hur (PMID: 34815573, 31626748) and Michaela Gack (PMID: 32203325, 36113429). In fact, PMID: 36113429 shows that actin cytoskeleton rearrangement can activate RIG-I. Therefore, I suggest incorporating some of these findings in the discussion and leave open the possibility of alternative interpretation of the reported results.

Referee #2:

The authors have addressed all my comments and I recommend this study for publication in EMBO J. There are a few minor issues the authors should clarify in the final files.

1. Comparing Fig 6G and EV3E, the requirement for MAVS is absolute when SKIV2L is deleted, whereas the requirement for RIG-I is partial (compare bar nb. 8 in both panels). Does this mean MDA5 is involved in sensing RNase L processed RNA in the setting of OAS GoF variant overexpression?
2. Comparing Fig 3E and EV2D, the requirement for MAVS is partial when SKIV2L is deleted, whereas the requirement for RIG-I is absolute (compare bar nb. 8 in 6G and bar nb. 6 in EV2D). This is confusing. According to the model in Fig 3F, there should be no response in MAVS KO cells. Please explain in text.
3. Fig 3E has a bracket on the left labelled 'MAVS dependent' and a bracket on the right labelled 'MAVS independent'. This is confusing.

Referee #3:

The authors have done an excellent job in addressing all questions that were raised by me or the other reviewers. The manuscript has substantially improved by 1) the addition of a rescue experiment, 2) the thorough validation of KO cell lines, 3) excluding the possibility that the observed phenotypes are due to increased priming/tonic IFN signaling, and 4) the addition of the patient fibroblasts to their current study. I therefore have no further comments. For clarity purposes, a small suggestion is that the authors note in the text or in the legend that the MAVS-independent IFN response in Fig. 3E is likely TLR3 dependent.

We would like to express our sincere gratitude to the editor and reviewers for their thoughtful and encouraging comments on our manuscript. We greatly appreciate the time and effort invested in reviewing our work. We have revised the manuscript to address the editorial comments and have incorporated the reviewers' suggestions. The text changes in the manuscript are highlighted in blue. Below, we provide a detailed point-by-point response to each comment.

Editorial comments

- The Figures should be removed from the main manuscript file and only be uploaded to our manuscript tracking system separately as individual Figure files. The legends of main and EV Figures should remain in the manuscript, after the list of References, with the headings "Figure legends" and "EV Figure legends", as appropriate.

Response: Done.

- Please include a list of up to 5 keywords after the Abstract of your revised manuscript.

Response: Added.

- The Materials and Methods need to be described in the manuscript using our "Structured Methods" format, which is now required for all research articles. According to this format, the Materials and Methods section includes a single "Reagents and Tools Table" -listing key reagents, experimental models, software and relevant equipment and including their sources and relevant identifiers- followed by a "Methods and Protocols" section describing the methods using a step-by-step protocol format. The aim is to facilitate adoption of the methodologies across labs. More information on this format as well as detailed instructions, examples, and a template (.docx) for the "Reagents and Tools Table" can be found in our author guide:

<https://www.embopress.org/page/journal/14602075/authorguide#structuredmethods>.

Response: Restructured.

- Please note that a "Data availability" statement (after the Materials and Methods section) is mandatory. If your study does not include newly generated datasets that need to be deposited in an external repository, please add the statement "This study includes no data deposited in external repositories." after the heading "Data availability".

Response: Added.

- Please note that a "Disclosure and competing interests statement" (after the "Data availability" section) is mandatory. Please find more information in our author guide:

<https://www.embopress.org/page/journal/14602075/authorguide#conflictsofinterest>.

Response: Added.

- The author contributions statement should be removed from the manuscript file. Instead, we now use CRediT to specify the contributions of each author in the journal submission system. Please feel free to use the free text box to provide more detailed descriptions during submission. See also our guide to authors for more information:

<https://www.embopress.org/page/journal/14602075/authorguide#authorshipguidelines>.

Response: Done.

- There is a callout for Figure S1A (on page 9 of the manuscript), but such a Figure does not exist. Please check and correct as appropriate.

Response: Fixed.

- Please include in your resubmission a completed Author Checklist, which you can download from our author guidelines (<https://www.embopress.org/page/journal/14602075/authorguide>). Please note that the checklist will also be part of the Peer Review File (see below for more information).

Response: Done.

- Please combine the Source Data for all EV Figures in a single master (zip) folder named "EV Figure Source Data".

Response: Done.

- Please note that EMBO press papers are accompanied online by:

A) a short (2 sentences) summary of the findings and their significance,

B) 2-5 short bullet points highlighting the key results, and

C) a synopsis image in .jpg or .png format that is exactly 550 pixels wide and 300-600 pixels high (the height is variable). Please note that the text needs to be legible at the final size. Please upload this information along with your revised manuscript (the text for A and B should be provided in a separate Word file).

Response: Uploaded.

- Please define the annotated p values ***/** as well as provide the exact p-values for the same in the legends of Figures EV 1h; EV 2c; EV 3d as appropriate.

Response: Added.

- Please note that the exact p values are not provided in the legends of Figures 1b, d, f, i, k; 2c; 3d-e; 5a-c; 6f-g; EV 1a-b; EV 2d; EV 3e.

Response: Added.

- Please indicate the statistical test used for data analysis in the legends of Figures EV 1h; EV 2c; EV 3d.

Response: Added.

- Please note that in Figure 4c there is a mismatch between the annotated p values in the Figure legend and the annotated p values in the Figure file that should be corrected.

Response: Corrected.

- Please note that information related to "n" is missing in the legends of Figures 1b, d, f, i, k.

Response: Added.

Referee #1:

In this revised manuscript the authors have addressed some of my concerns and added substantial new results. Therefore, I'm supportive of its publication in EMBO J. However, some of the explanations provided by the authors needs further discussion. For example, in response to my previous comment #2 the authors states: "After poly(I:C) stimulation, we observed no major difference

in the activation of the PKR-eIF2a pathway (measured by phosphorylation of PKR and eIF2a) between WT and SKIV2L KO cells (Fig. 2A and Fig. 3A lane 5 versus lane 6). These data suggest that SKIV2L RNA exosome does not substantially degrade incoming external dsRNA, as such degradation would likely impair the PKR-eIF2 α RNA sensing pathway." But, the nature of RNA ligands that activate PKR (dsRNA of certain length) vs RLR pathways (ds and structured ssRNA with 5' phosphates) are quite different. Thus, similar PKR activity doesn't really answer this question.

Response: We acknowledge the structural differences between synthetic dsRNA analog poly(I:C) and natural RNA ligands, such as those derived from viruses. One previous study in *Drosophila* cells has shown that the *nuclear* RNA exosome can target viral RNA for degradation (PMID: 27474443). However, it remains unclear whether the mammalian *cytoplasmic* SKIV2L RNA exosome exhibits a similar capacity to degrade viral RNA. In the revised manuscript, we have discussed the limitations of using synthetic dsRNA analogs like poly(I:C) in our study and have cautioned against over-interpreting our findings in the context of viral infections.

In #4 the authors used RNA isolated from human A549 cells expressing OAS-GoF mutant to naturally activate RNase L without introducing external dsRNA, then isolated cellular RNA from these cells and transfected it into a wild-type MEF cells showing the RNA from SKIV2L-KO exhibited increased immunostimulatory activity, consistent with the proposed model. However, the reason for why these processed RNA don't activate RLR signaling in the same cell, and requires transfection is not elaborated.

Response: Sorry for the confusion. When OAS1-GoF was induced in *SKIV2L*^{KO} A549 cells, we **did** observe increased IFN response in the cell (**Fig. 6E**) and further deletion of *MAVS* or *DHX58* reduced IFN response (**Fig. 6G and Fig. EV3E**). To test whether the increased IFN response is due to increased immunostimulatory RNA in *SKIV2L*^{KO} cells, we transfected RNA isolated from OAS-GoF-expressing A549 cell into recipient wild-type MEFs and again observed increased IFN response, confirming the presence of immunostimulatory RNA.

Another point is the nature of RLR activation and the ligand necessary have been studied extensively, well beyond PMID: 17653195. Please see studies from Sun Hur (PMID: 34815573, 31626748) and Michaela Gack (PMID: 32203325, 36113429). In fact, PMID: 36113429 shows that actin cytoskeleton rearrangement can activate RIG-I. Therefore, I suggest incorporating some of these findings in the discussion and leave open the possibility of alternative interpretation of the reported results.

Response: We appreciate the reviewer's suggestion and have added these findings into the discussion.

Referee #2:

The authors have addressed all my comments and I recommend this study for publication in EMBO J. There are a few minor issues the authors should clarify in the final files.

1. Comparing Fig 6G and EV3E, the requirement for MAVS is absolute when SKIV2L is deleted, whereas the requirement for RIG-I is partial (compare bar nb. 8 in both panels). Does this mean MDA5 is involved in sensing RNase L processed RNA in the setting of OAS GoF variant overexpression?

Response: Thank you for this important observation. We noticed the difference between MAVS KO and RIG-I KO. The results suggest that, in the setting of OAS1 GoF, RNase L processing may generate endogenous RNA ligands for both RIG-I and MDA5. Further investigation is needed to elucidate the biochemical nature of these RNAs.

2. Comparing Fig 3E and EV2D, the requirement for MAVS is partial when SKIV2L is deleted, whereas the requirement for RIG-I is absolute (compare bar nb. 8 in 6G and bar nb. 6 in EV2D). This is confusing. According to the model in Fig 3F, there should be no response in MAVS KO cells. Please explain in text.

Response: We observed that MAVS KO and RIG-I KO A549 cells actually still responded to poly(I:C) to a less extent (see Fig. 3E and Fig. EV2D below, zoomed in on the lower range of the y-axis). Previous studies have shown that, in addition to RIG-I-like receptors, A549 cells also express the dsRNA receptor TLR3, which mediates IFN response after viral infection or poly(I:C) treatment (PMID: 25880109, 19234180, 15778392, 15731229). Therefore, we speculate that the residual IFN response is attributed to RLR-MAVS-independent RNA sensing pathway, such as TLR3. Another difference between Fig. 3E and Fig. EV2D is that higher concentration of poly(I:C) was used in Fig. 3E (note y-axis "fold 1000x"), which may explain why we observed more MAVS-independent IFN response in Fig. 3E. In the revised manuscript, we noted MAVS-independent IFN response in the poly(I:C)-stimulated A549 cells to clarify the residual IFN response.

3. Fig 3E has a bracket on the left labelled 'MAVS dependent' and a bracket on the right labelled 'MAVS independent'. This is confusing.

Response: Thanks for bringing this up. The left 'MAVS dependent' bracket was meant to denote the difference in IFN response between WT and MAVS^{KO}, and the right labelling was for comparison between SKIV2L^{KO} and SKIV2L^{KO}MAVS^{KO}. To avoid confusion, we removed the left one in revision.

Referee #3:

The authors have done an excellent job in addressing all questions that were raised by me or the other reviewers. The manuscript has substantially improved by 1) the addition of a rescue experiment, 2) the thorough validation of KO cell lines, 3) excluding the possibility that the observed phenotypes are due to increased priming/tonic IFN signaling, and 4) the addition of the patient fibroblasts to their current study. I therefore have no further comments. For clarity purposes, a small suggestion is that the authors note in the text or in the legend that the MAVS-independent IFN response in Fig. 3E is likely TLR3 dependent.

Response: We appreciate the reviewer's positive remarks. We noted the MAVS-independent IFN response is likely TLR3 dependent in the revised manuscript.

Dear Nan,

Congratulations on an excellent manuscript, I am very pleased to inform you that it has been accepted for publication in The EMBO Journal. Thank you for your comprehensive responses to the referee concerns and editorial requests.

If you have any questions, please do not hesitate to contact the Editorial Office. Thank you for your contribution to The EMBO Journal. Working with you has been a pleasure!

Best wishes,

Ioannis
